# The C-terminal tail of α-synuclein protects against aggregate replication but is critical for oligomerization

Azad Farzadfard[1,2], Jannik Nedergaard Pedersen[1], Georg Meisl [3], Arun Kumar Somavarapu[1], Parvez Alam[1,4], Louise Goksøyr [5], Morten Agertoug Nielsen [5], Adam Frederik Sander [5], Tuomas P. J. Knowles [3,6], Jan Skov Pedersen [1,7] & Daniel Erik Otzen [1,8]✉

Aggregation of the 140-residue protein α-synuclein (αSN) is a key factor in the etiology of Parkinson's disease. Although the intensely anionic C-terminal domain (CTD) of αSN does not form part of the amyloid core region or affect membrane binding ability, truncation or reduction of charges in the CTD promotes fibrillation through as yet unknown mechanisms. Here, we study stepwise truncated CTDs and identify a threshold region around residue 121; constructs shorter than this dramatically increase their fibrillation tendency. Remarkably, these effects persist even when as little as 10% of the truncated variant is mixed with the full-length protein. Increased fibrillation can be explained by a substantial increase in self-replication, most likely via fragmentation. Paradoxically, truncation also suppresses toxic oligomer formation, and oligomers that can be formed by chemical modification show reduced membrane affinity and cytotoxicity. These remarkable changes correlate to the loss of negative electrostatic potential in the CTD and highlight a double-edged electrostatic safety guard.

[1] Interdisciplinary Nanoscience Center (iNANO), Aarhus University, Gustav Wieds Vej 14, 8000 Aarhus C, Denmark. [2] School of Biology, College of Science, University of Tehran, Tehran, Iran. [3] Yusuf Hamied Department of Chemistry, University of Cambridge, Lensfield Road, Cambridge CB2 1EW, UK. [4] Department of Biomedicine, Aarhus University, 8000 Aarhus C, Denmark. [5] Centre for Medical Parasitology at the Department of Immunology and Microbiology, University of Copenhagen, Blegdamsvej 3B, 2200 Copenhagen N, Denmark. [6] Cavendish Laboratory, University of Cambridge, J J Thomson Avenue, Cambridge CB3 0HE, UK. [7] Department of Chemistry, Aarhus University, Langelandsgade 140, 8000 Aarhus C, Denmark. [8] Department of Molecular Biology and Genetics, Gustav Wieds Vej 10C, 8000 Aarhus C, Denmark. ✉email: dao@inano.au.dk

Alpha-synuclein (αSN) is a 140-residue intrinsically disordered protein, mainly expressed in brain cells[1]. Aggregation of αSN to form insoluble fibrils and smaller oligomeric species (αSOs), a process which is increased in different familial variants of Parkinson's disease (PD)[2], is critical to the onset and development of PD. This involves neuronal toxicity propagated through the brain, presumably via the transmission of αSN aggregates from cell to cell[1]. Thus, control of αSN aggregation and transmission may eventually lead to therapies against PD.

αSN can be divided into 3 regions. The N-terminal domain (NTD, residues 1–60) mediates binding to synaptic vesicles, which is believed to contribute to αSO toxicity[3,4]. The NAC domain (residues 61–95) forms the core folded region of both αSOs[5] and fibrils[6,7]. Finally, the highly negatively charged C-terminal domain (CTD, residues 96–140), which remains mobile both in αSOs[5,8] and fibrils[6,7], can also bind to synaptic vesicles through bridging $Ca^{2+}$ ions[9,10]. The CTD's high mobility and accessibility in all αSN species makes CTD truncation of αSN a relatively common occurrence in vivo. Truncations at residues 103, 110, 113, 114, 115, 119, 122, 124, 125, 133, and 135 have been identified in vivo[11]. Those at 119 and 122 at most prevalent and occur at 20–25% abundance relative to full-length αSN. The truncations typically result from incomplete degradation of full-length αSN by the proteasome[12] or other proteases like Calpain I[13]. CTD truncation is linked to PD, though the exact causality is unclear. CTD truncation produced by alternative splicing in response to oxidative stress results in proteosomal dysfunction and cell death[14]. Transgenic mice and cells expressing αSN mutants linked to familial PD produce CTD-truncated αSN[15]. Co-expression of CTD-truncated αSN in rats induces accumulation and aggregation of full-length αSN and impaired dopamine release[16]. CTD-specific antibodies ameliorate neurodegeneration pathology and behavior in PD-mice model and inhibit disease propagation[17].

Besides these compelling cellular data, CTD truncation accelerates αSN fibrillation in vitro by decreasing the fibrillation lag time and increasing the extent of fibrillation[18–21]. The effect is proportional to the removal of negative charge, indicating that full-length CTD may suppress aggregation through long-range interactions with the NAC[22] or N-terminal region[23]. Fibrillation is accelerated by screening the CTD's negative charges through cations[20] or lowered pH[19]. CTD truncation also leads to twisted, thinner, and shorter fragmented fibrils[21,24,25]. This may reflect decreased charge repulsion between individual fibrils and increased exposure of hydrophobic surface area and intermolecular β-sheets[23].

αSOs likely exert toxicity by vesicle binding, leading to synaptic impairment and neuronal dysfunction[26–28]. Studies are challenged by many types of reported αSOs[29]. However, the most prevalent species (formed in PBS buffer at 37 °C and high concentrations of αSN) assumes an ellipsoidal core-shell structure, in which a compact β-sheet rich core (the NAC region and part of the NTD) is surrounded by a shell of dynamic residues (mostly the CTD[5])[29–32]. This species is highly stable[33] and weakly blocks αSN fibrillation[32], indicating that it is an off-pathway species. Similar αSOs form with PD-related lipid oxidation products such as 4-oxo-2-nonenal (ONE)[34]. Despite its dynamic nature, the CTD also plays an important role in the formation of αSOs. For instance, its interactions with $Cu^{2+}$ and $Ca^{2+}$ induces αSO formation[35,36]. The neurotransmitter dopamine inhibits αSN aggregation and stabilizes αSOs by interacting with the motif $Y_{125}EMPS_{129}$ in the CTD[37,38].

It remains unclear how CTD truncations affect the molecular mechanism of fibrillation and the formation and properties of oligomeric species. Using a series of truncation mutants we show that removal of more than 15 residues strongly enhances fibrillation thanks to increased rate of replication via a secondary process (fragmentation or secondary nucleation). Remarkably, oligomerization is essentially abolished unless ONE-assisted. These truncated oligomers are however larger, less stable, and much less disruptive towards vesicles and cells. Thus, the C-terminal tail plays a central role in αSN structure and function.

## Results

**C-terminal truncation accelerates secondary processes in αSN fibrillation.** To correlate changes in self-assembly with C-terminal tail length, 9 different αSN constructs were designed in which the CTD was truncated in steps of 4–6 residues from full length αSN to residue 100. αSN-FL is full-length αSN while e.g., αSN115 refers to the construct consisting of the first 115 residues of αSN (and thus lacking residues 116–140). Recombinant expression levels in *E. coli* diminished steeply for constructs αSN110 and shorter, declining to ~20% (compared to full-length) for αSN100 (Fig. S1).

To probe how truncations alter αSN aggregation, we recorded the kinetics of aggregation of each construct at a range of monomer concentration ([monomer]) (Figs. S2 and S3) and compared with integrated rate laws using the web-based program Amylofit[39] which allows to identify kinetic models based on microscopic mechanisms that best describe macroscopic aggregation behavior. SDS-PAGE analyses confirmed that the great majority of αSN had aggregated by the end of the reaction (Fig. S4). (Note that slightly incomplete levels of fibrillation will not affect the outcome of the analysis in Amylofit.) TEM demonstrated very similar architecture for the fibrils formed by the different truncation variants (Fig. S5). In a plot of the half-time of fibrillation versus [monomer] on a double logarithmic plot (Fig. 1), the slope provides the scaling exponent γ, which serves as a guide to the dominant mechanism of aggregate formation. A single linear trend implies that the same process dominates over the whole concentration range, while a change in slope implies a change in the rate determining mechanism[40]. Strikingly, the constructs fall into two groups. Group one (αSN125 up to αSN-FL) shows a single linear correlation over the whole concentration range. In group two (shorter than αSN125), a second linear regime emerges at high concentrations with a flat slope, i.e., fibrillation half times become independent of [monomer]. This starts to become visible for αSN121 and is very prominent for αSN105 (αSN100 aggregated very rapidly and irreproducibly). This implies that the fibrillation mechanism changes at high concentrations for the shorter constructs, most likely due to saturation effects of elongation or primary nucleation, which decrease the [monomer] dependence of the rate determining steps[40].

The fibrillation curves for the group one constructs are best described by a model in which fibrils elongate by monomer addition in a process which is first order in [monomer], and multiply by a secondary process independent of [monomer] and reaction order zero, such as fragmentation or saturated secondary nucleation, the latter consistent with previous observations[41]. Fits with different molecular aggregation mechanisms are shown for αSN-FL in Fig. S2 along with plots fitted to models with secondary processes for αSN135-αSN125. For group two constructs, comparison of this model has to be extended to allow for saturation of the elongation process (Fig. S3). Considering the elongation process as a two-step reaction (i.e., binding of monomers to the end of fibrils and their conformational conversion to fibrils), saturation of elongation implies that at higher [monomer], the reaction no longer exhibits first order kinetics with respect to [monomer] and the rate-limiting step

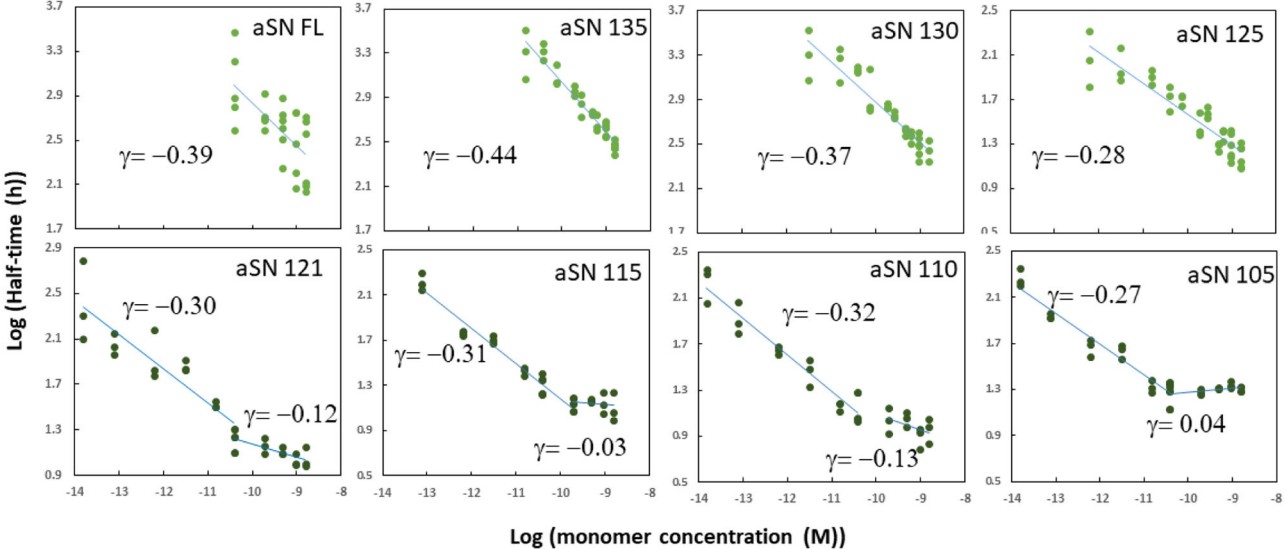

**Fig. 1 Effect of C-terminal truncation on αSN fibrillation.** Log-log plots of fibrillation half-times versus concentrations whose slope provides the scaling exponent γ. The plots reveal a single linear correlation for αSN125 and longer constructs and the growing prominence of a flat region at high protein concentration for shorter constructs. Each point represents the half-time of a ThT curve (i.e., the time to reach half the maximal fluorescence value) from a fibrillation experiment carried out at the indicated αSN concentration.

shifts from monomer binding to the end of fibrils (which is dependent on [monomer]) to the conformational conversion of the monomer to the fibrillated state (which is not). These limiting cases correspond to the two linear regimes in Fig. 1.

As shown in Figs. S2 and S3, C-terminal truncation of αSN accelerates the fibrillation both by decreasing the lag time and by increasing the slope of the sigmoidal aggregation curves. We used global kinetic analysis to elucidate the mechanistic basis for this behavior. To compare rates of individual aggregation processes for different constructs, data for each construct were fitted with a model that includes a saturating elongation and a monomer-independent secondary process model. The rate constants describing primary nucleation ($k_n$), elongation ($k_+$) and secondary processes (fragmentation $k_-$ or secondary nucleation $k_2$) are obtained as combinations (e.g., $k_n*k_+$ and $k_-*k_+$) from global fits of aggregation data in the absence of preformed seeds. To decompose the observed changes in these combined rate constants into changes in individual molecular rate constants, we determined elongation rates for the constructs using a fibrillation assay in the presence of high and variable amounts (5–25%) of seeds with 50 μM monomeric αSN. Under such conditions, the integrated rate laws exhibit a direct dependence on $k_+$. The relative elongation rate constants for all constructs (from plots of the initial rate of fibrillation against the concentration of seed (Fig. S6) are all very similar and do not show any systematic variation with truncation (Fig. 2a) at 50 μM monomer. We used these relative elongation rates together with the composite values to obtain relative rate constants of primary nucleation and of the secondary process (in both cases normalized against the value for αSN-FL). To compare different constructs with different reaction orders of nucleation, we calculated the nucleation rate at 50 μM, which is in the middle of the concentration range where the relative elongation rate was measured experimentally. There is a weak trend for primary nucleation rates to fall with construct length (Fig. 2b); more strikingly, we see a jump between two relatively constant levels in the rate of the secondary process (Fig. 2c) between αSN130 and αSN125 (we attribute the low $k_n$ value of αSN115 to a fitting artifact in which a low $k_n$ value is compensated by a higher $k_-$ value). The increase in second rate constant reflects the sudden

increase in the slopes of the sigmoidal curves between αSN130 and αSN125 (which the more noisy variation in primary nucleation rates does not) and demonstrates that the secondary process drives accelerated fibrillation in αSN125 to αSN105 (Fig. 2c).

This secondary process could either be fragmentation of growing fibrils or surface-catalyzed secondary nucleation (Fig. 2d). Distinguishing between saturated secondary nucleation and fragmentation can often be difficult. The scaling exponent γ obtained from the log plot of fibrillation half-time versus [monomer][39] (Fig. 1) is comparable to −0.5, the value obtained from theory for a system with fragmentation or saturated secondary nucleation, for all mutants[39]. We therefore compared αSN fibrillation in the presence of low concentrations of seeds, one under quiescent conditions and the other with shaking. Shaking has been shown to promote primary nucleation, by increasing the air-water interface, as well as fibril fragmentation[42–44]. Secondary nucleation is expected to be less dependent on shaking, though one could also envision an increase in the turnover rate, i.e., the rate of detachment of formed nuclei from the surface of fibrils and attachment of fresh monomer induced by agitation leading to an increase in the secondary nucleation rate. Our results are very clear: in the quiescent mode experiment, no ThT binding signal was observed after 96 h, whereas fibrillation was completed after few hours in shaking mode for the truncated mutants and FL (Fig. 2e, f shows data for αSN121 while Fig. S7 shows the same behavior for other mutants as well as αSN FL). The fact that very low concentrations of seeds (0.03%) strongly promoted the fibrillation demonstrates the dominance of secondary processes which, as outlined above, we believe is most likely agitation- induced fragmentation.

**The accelerating effect persists with low levels of truncated αSN.** To investigate the effect of co-incubating full-length and truncated αSN, we fibrillated a mixture of αSN-FL and αSN110 at constant total concentration but variable proportions of the two proteins (Fig. 2g). Remarkably, as little as 10% αSN110 shifted the fibrillation time course away from the slower fibrillating αSN-FL and closer to that of the faster fibrillating αSN110. The effect

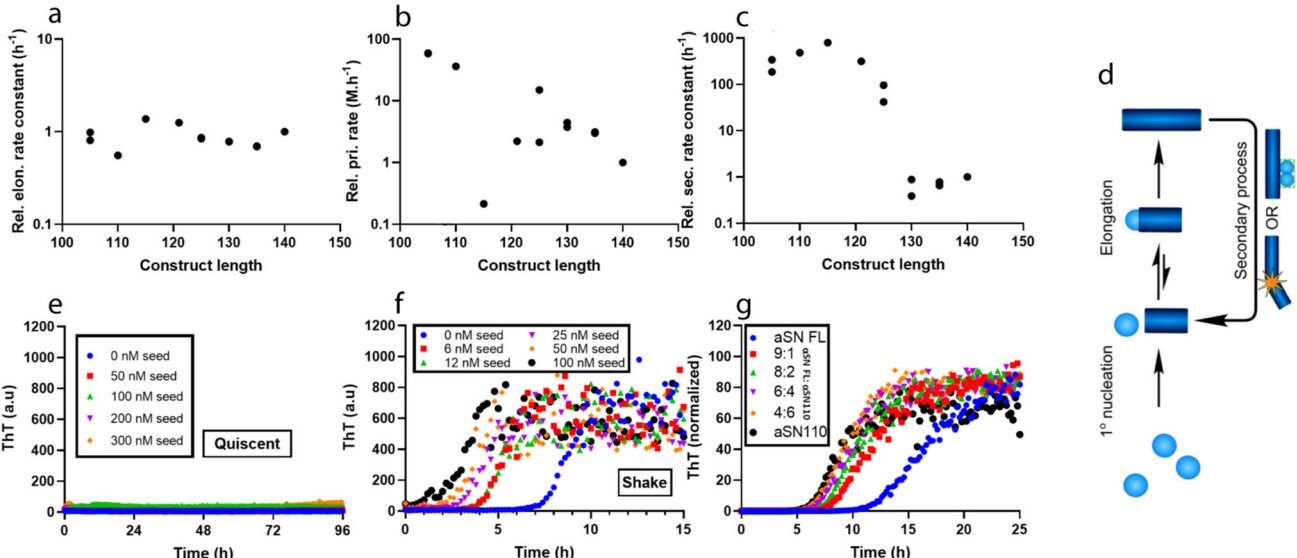

**Fig. 2 Determination of individual rate constants of fibrillation for αSN truncated mutants. a** Relative elongation rate constant, $k_{+, rel}$, obtained from seeded experiments shown in Fig. S6. There is little variation with length under the assumption that the length of the seed fibrils is similar for different truncation lengths. Details in Eq. 3. **b** The rate of primary nucleation, evaluated from the rate constant, $k_n$, and reaction order, $n_c$. Values calculated at 50 μM αSN to enable comparison between variants with varying reaction orders. **c** Rate constant of fragmentation, $k_-$, versus construct length. There is a very abrupt decrease between αSN125 and αSN130. Each data point in **a**–**c** corresponds to a separate experiment with three repeats in three wells; αSN 105, αSN 125, aSN 130, and αSN 135 were randomly repeated. **d** Schematic reaction network with an explicit two-step mechanism for elongation. **e, f** Low-seed (0.03, 0.06, 0.1, 0.2%) fibrillation without (**e**) and with (**f**) shaking shows that shaking strongly promotes fibrillation. Data are for αSN121. (**g**) ThT fibrillation of different mixtures of αSN-FL and αSN110 (total protein concentration of 100 μM). Note that as little as 10% αSN110 significantly reduces the lag time. Shaking conditions in **f, g** are reduced to 1 min (instead of the 10 min used for the other shaking experiments) in 12-min measurement cycles.

saturated quickly, and 60% αSN110 showed the same behavior as 100% αSN110. If truncated αSN mutants are more prone to fragmentation, then incorporation of even small amounts of these variants into a mixed fibril may disproportionately increase the rate of fragmentation. If instead truncation increases secondary nucleation, then the increased rate in the mixture may be due to the increased availability of defects and other catalytic sites on the fibril surface, believed to promote secondary nucleation. These results demonstrate the key importance of accounting for fast aggregating truncated versions, even when the full-length protein is the predominant form and truncated versions are present only in small proportions, as may be the case in vivo[15,16].

**Shorter constructs make bigger and longer αSOs which are less disruptive towards vesicles.** To test the general tendency of CTD truncation mutants to form αSOs, we first optimized oligomerization conditions for αSN-FL by exploring parameters such as NaCl concentration, pH, shaking, multiple rounds of freeze-drying as well as the αSO inducing agent 4-oxononenal (ONE). Among these, multiple rounds of freeze-drying[45] and ONE modification[34] were most effective. αSOs accumulated in a linear fashion with each cycle of freeze-drying (Fig. S8a, b) without any change in structure according to Small Angle X-ray Scattering (Fig. S8c). Changes in [NaCl] did not help, but increasing pH to 10 increased yields by ca. 30% compared to pH 7.4, while oligomerization completely stopped around and below pH 6 (Fig. S8d–f). The only residue expected to titrate around pH 6–7 in αSN is the single His residue in position 50 (pK$_a$ ~ 6). Remarkably, the mutant His50Ala failed to oligomerize over a broad pH range, even after 3 cycles of freeze-drying (Fig. S8g).

Strikingly, while αSN121 and longer all gave the same yield of αSOs, based on the oligomer/monomer peaks height ratio (Fig. S8h, i), shorter constructs did not form any αSOs at all but remained monomeric even after three rounds of freeze-drying (Fig. S8h). These shorter constructs only formed αSOs with ONE

(Fig. S8j), leading to comparable yields of the main component oligomer (eluting around 12 ml) in all cases.

We investigated changes in shape and size of truncated αSOs by SAXS according to the previously developed model for αSN WT oligomer[32]. The core-shell model fitted both unmodified and ONE-modified truncated αSOs well (Fig. S9a, b). Pair distance distribution function ($p(r)$) curves showed an increase in maximal $r$-values with truncation, particularly for αSOs of length 115 and shorter (Fig. S9c, d). Correspondingly, truncation slightly increases radius of gyration in unmodified αSOs (R$_g$) (Fig. 3a). In ONE-modified αSOs, changes in the long axis, shell thickness and R$_g$ (Fig. 3b) cluster in two groups: group one similar to ONE-αSO-FL (ONE-αSO121 and longer) and group two with much bigger αSOs (shorter than αSO121). Again, truncation around 120 residues is critical. Group two consists of constructs unable to form αSO without ONE. The number of amino acid residues in the shell of ONE-modified αSOs decreased with truncation, while there was no clear trend for unmodified αSOs (Fig. 3c) (a trend towards more residues per shell was obscured by αSO121). For all αSOs, truncation increased the number of monomers per αSO (Fig. 3d).

There was only a very modest overall reduction in β-sheet content in ONE-modified αSOs (ONE-αSO) compared to unmodified αSOs (from 39 to 32%), and the truncations do not significantly change αSO secondary structure (Fig. S9e–h). TEM images showed elongated ONE-modified αSOs but ring-shaped unmodified αSOs (Fig. S10). However, the shapes of truncated αSOs were similar to that of αSN-FL, and both αSOs and ONE-modified αSOs had diameters of ~10 nm. Oligomer size measured by TEM is similarly ~10 nm for all the mutants, consistent with the core region being stained, while the more dynamic shell region remains unstained.

There is substantial evidence that permeabilization of cell membranes by αSOs leads to calcium ion influx and cell death[46–48]. As a simple model system to mimic this phenomenon,

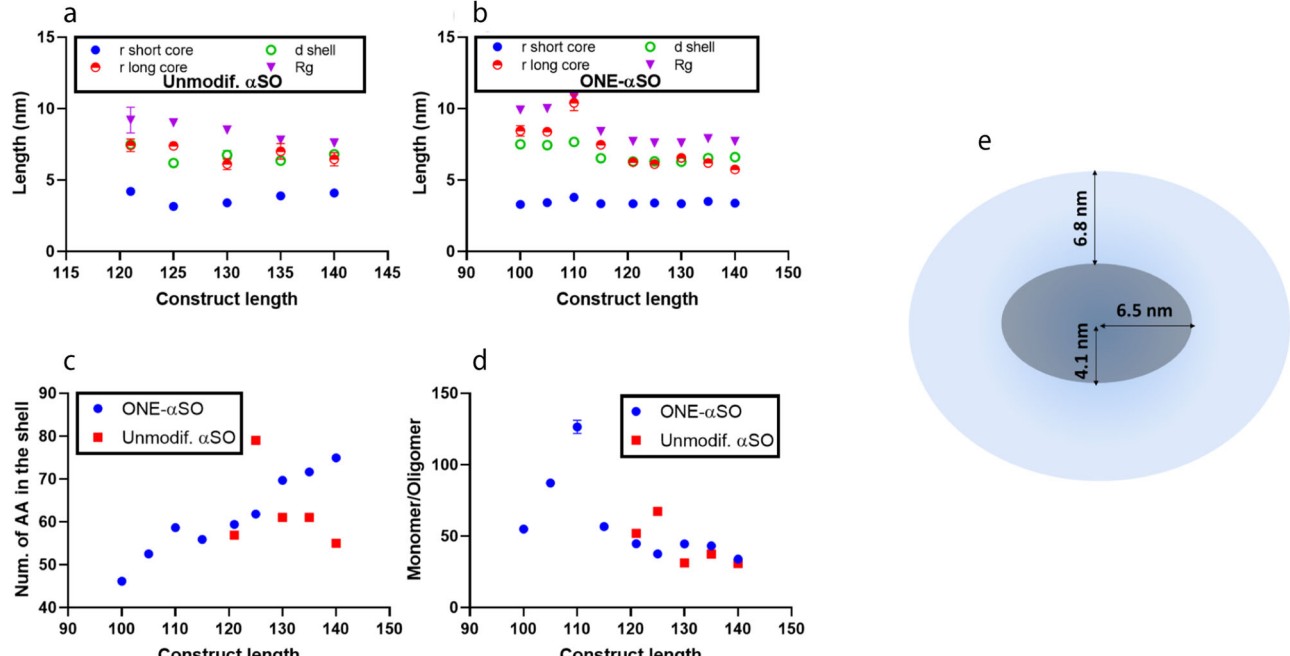

**Fig. 3 Properties of ONE-modified and unmodified oligomers of truncated αSN mutants. a–d** SAXS analysis of C-terminal truncated αSN oligomers. **a**, **b** The data with a previously developed model for αSO140 suggest similar ellipsoidal shapes for all oligomers, i.e., a dense core and diffuse shell. However, parameters related to the shape of core and shell change in different ways in truncated constructs of unmodified versus ONE-modified αSOs. **c** The number of amino acids in the shell fluctuate in unmodified αSOs but decrease with truncation in ONE-modified αSOs. **d** There is also an increase in the number of monomers per oligomer for both kinds of oligomers. Each point is the average of three replicates where error bars show standard deviations. **e** Schematic model of unmodified αSN FL oligomers based on SAXS[32]. A compact core is surrounded by a shell of more flexible polypeptide chains.

we determined the ability of these αSO constructs to permeabilize negatively charged vesicles, quantified as the concentration of protein needed for 50% calcein release (CR[50%]). Though all truncated αSN monomers were disruptive, CR[50%] values increased significantly with truncation (Fig. 4a, b). Like their monomeric counterparts, truncation decreases αSO disruptiveness (Fig. 4c, d) but αSOs were more vesicle-disruptive than monomers. Strikingly, although calcein release from the longer ONE-modified αSOs (ONE-αSO125 and longer) was similar to unmodified αSOs, ONE-modified αSOs of shorter constructs (ONE-αSO100 to ONE-αSO121) showed reduced disruption (Fig. 4e, f). Gratifyingly, these results agreed with αSO cell toxicity assays on SH-SY5Y neuroblastoma cells, exposed for 24 h to αSOs (Fig. 4g). With the exclusion of αSO121, there is a good correlation between CR[50%] values and cell viability (Fig. S11), supporting calcein release as a proxy for cell toxicity. Toxicity of unmodified αSOs was unaffected by truncation, whereas ONE-modified αSOs divide with statistical significance into longer αSOs with higher toxicity and shorter αSOs with lower toxicity (Fig. 4h).

## Discussion

C-terminal truncation of αSN led to two conspicuous changes in fibrillation kinetics. Firstly, a saturation of the elongation step for αSN121 and shorter constructs. Secondly, a steep increase in the rate of the secondary process for αSN125 and shorter constructs led to faster fibrillation of truncated αSN. An increased rate of the secondary process, either through an increased tendency to fragment or (less likely) through an increased secondary nucleation rate, should lead to shorter average lengths of fibrils. Consistent with increased fragmentation, shorter C-terminal truncated constructs of αSN are known to produce shorter fibrils[21].

**Critical role of the C-terminal 20 residues of αSN in modulating intramolecular contacts through electrostatic interactions.** The effect of CTD on fibrillation is well-documented. Truncation of CTD at positions 103, 118, and 129 or replacing anionic residues in CTD with neutral counterparts speeds up fibrillation[49]. CTD's protective role against fibrillation has been attributed to the interaction of the negatively charged CTD with the positively charged N-terminal and NAC regions[22,23,50]. Hydrogen exchange (HX) of the αSN monomer shows protection in the CTD that indicates transient long range interaction between CTD and NTD[51]. These interactions can be screened by Ca$^{2+}$ ions that bind to CTD and increase solvent accessibility of NTD, resulting in a higher aggregation propensity[50]. A systematic analysis of the electrostatic repercussions of truncation is illuminating. The electrostatic potential of αSN remains essentially zero for the first 100 residues, after which it rises to a plateau from residue 120 onward[52]. Together with our data, these findings indicate a critical role for the C-terminal 20 residues, which likely interact with the NTD to regulate αSN fibrillation and oligomer formation. Removal of the residues with higher electrostatic potential frees up the NTD/NAC region for intermolecular contacts. In support of this, NAC-derived peptides bind to the CTD and accelerate aggregation, probably by blocking NAC-CTD contacts[53]. As further evidence that CTD contacts with NTD/NAC can block fibrillation, mutants containing intramolecular disulfide bonds between residues 107 and 124 in CTD combined with intramolecular disulfide bonds in the NTD-NAC region (residues 9, 42, 69, and 89) do not fibrillate; by contrast disulfide bonds in the NTD-NAC region alone are insufficient to inhibit fibrillation[54].

**CTD truncation introduces a saturation step in elongation.** Faster elongation in CTD truncated constructs (caused by loss of

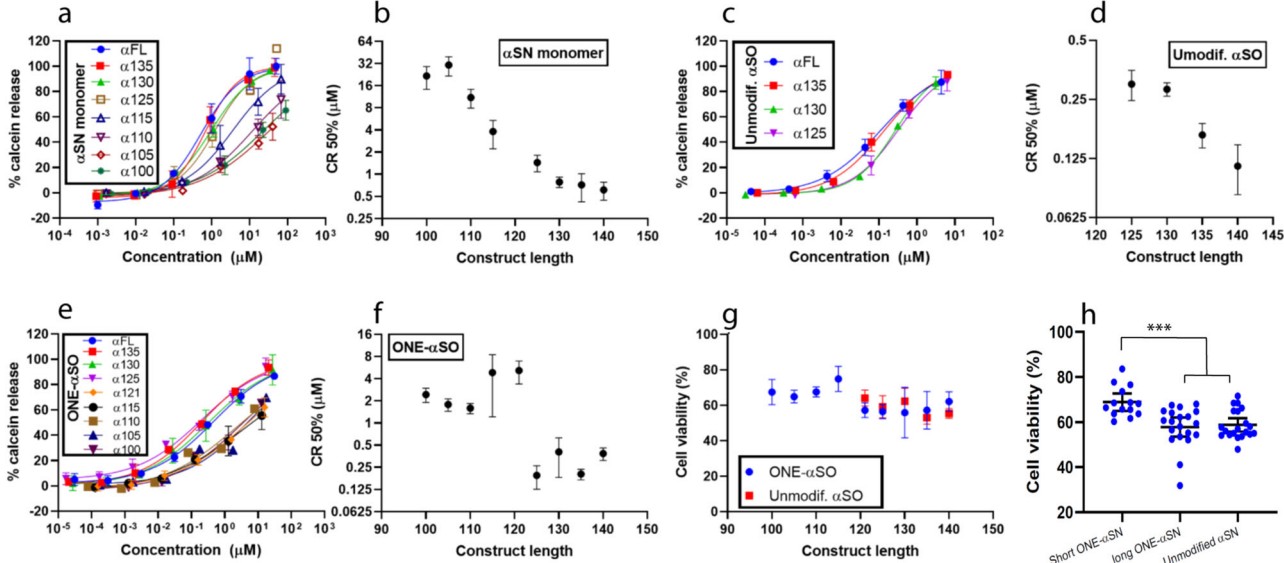

**Fig. 4 Vesicle permeabilization and cell toxicity properties of CTD truncated αSN and their oligomers. a–f** Vesicle permeabilization by truncated αSN constructs. Permeabilization assays with truncated αSN monomers (**a**, **b**), unmodified αSOs (**c**, **d**), and ONE-modified αSN monomers (**e**, **f**) shows more aggressive behavior for oligomers in comparison to monomers, however, ONE-αSO121 and the shorter ones are not as aggressive as the ONE-αSO125 and longer oligomers. **g**, **h** Cell viability with 10 μM oligomer confirms the results of vesicle permeabilization. In **h** αSOs of αSN121 and longer variants are binned into one group and αSN115 and shorter variants into another. Cell toxicity of each group was compared by ANOVA (in graph pad software). ***$p$-value ≤ 0.001. Each point is the average of three repeats for permeabilization or four repeats for cell toxicity where error bars show standard deviation. Error bars for **h** show 95% CI.

interactions with the NTD/NAC) is accompanied by saturation of elongation. Elongation can be described as a two-step process, with monomer attachment to the fibril end, followed by conversion into the fibril structure. A saturation phenomenon with increasing levels of truncation could arise either from changes in binding ($K_E$) or conversion ($k_c$). Despite the observed decrease in $K_E$ upon truncation, there is a slight decrease in the overall rate of elongation $k_+$. This implies that the conversion step ($k_c$) is slowed significantly by truncation.

Saturation of elongation of shorter constructs might be induced by a change in the structure of the formed fibril, leading to slower conversion. A CTD-truncated αSN lacking residues 109–140 produces more twisted fibrils with greater distance between the β-sheets and higher solvent exposure in comparison with αSN-FL fibers[24]. Moreover, CTD-truncated fibrils do not elongate αSN-FL in vitro and in cells to the same extent as αSN-FL fibrils do[21,24,55], supporting a change in fibril structure. In contrast, monomers of truncated αSN in a mixture with αSN-FL monomer accelerate fibrillation of αSN-FL in vitro and in cells[21]. We recapitulate this in our study, where 10% of CTD truncated mutant significantly accelerates αSN fibrillation. Even small amounts of truncated αSN significantly increase production of seeds to make new fibrils and propagate disease. Thus, as implied elsewhere[21], CTD truncated αSN monomers may promote spread of PD.

**CTD truncation and His50 protonation both compromise oligomerization.** Long-range interaction between the CTD and NAC/NTD domains of αSN might also be important for formation of αSN oligomers, since αSN121 and shorter mutants, which weaken or remove such interactions, fail to oligomerize. The inability to oligomerize at and below pH 6 points to a critical protonation around this pH value. The only candidate is the single His residue in αSN (His50). Remarkably, αSN H50A was unable to oligomerize between pH 6 and 9.5 (Fig. S8g), implying a critical role for a deprotonated imidazole ring in oligomerization. The basis for this remains unclear in the absence of

a high-resolution (or residue-specific) oligomer structure. We note that residue 50 is part of a region of αSO which forms relatively stable secondary structure according to hydrogen exchange MS[5], though the key structured region is centred around residues 70–88 according to solid state NMR[56].

It is noteworthy that ONE allows all αSN constructs to oligomerize. ONE is a product of peroxidation of polyunsaturated fatty acids present in the cell membrane[57], which has been reported to cross-link brain proteins[58]. In particular, ONE can induce αSN oligomer formation which could be physiologically relevant considering the role of oxidative stress in PD[34,59]. Therefore, we investigated ONE-induced oligomer formation in CTD truncated mutants that do not form oligomers on their own. ONE oligomers are more stable against dissociation in SDS or urea than normal oligomers[34]. ONE mostly modifies Lys residues and can thus crosslink different αSN molecules covalently[34,60]. ONE oligomers of shorter constructs are more elongated with relatively bigger shells than oligomers of aSN-FL (whether ONE modified or not). These oligomer properties illustrate how CTD truncation divides αSN constructs into two groups with different physiological properties. It is remarkable that ONE induces oligomer formation even in shorter constructs which are normally not able to form oligomer. Given that ONE-oligomers and unmodified oligomers have similar size and shape according to SAXS, we conclude that ONE helps otherwise unstable truncated oligomers form, instead of directing αSNs towards a different class of oligomers. In other words, ONE could be seen as filling in CTD's role in stabilizing the oligomer. The basis for this remains unclear. One speculative possibility is that just as the CTD's anionic tail may reduce the electrostatic repulsion caused by the more cationic N-terminal region, similarly ONE may overcome the N-terminal positive charges either by modifying the Lys side chains or by cross-linking them to other parts of the protein. If inter-molecular interactions between CTD and NTD in the oligomeric state indeed helps form and stabilize the oligomers, then conversely, shortening of the CTD will decrease

the frequency and strength of these interactions, leading to extended tails and therefore oligomers with bigger shells. However, there is a limit to the ability of ONE to compensate for truncation since removal of sufficient residues (around αSN115) completely abolishes the ability to form oligomers without help from ONE.

Finally, the properties of aSOs are reflected in their ability to disrupt membranes and kill cells. Although the membrane penetration of monomeric aSN decreases gradually as CTD is truncated, oligomers of these mutants again fall into two groups: aSOs of shorter constructs with less membrane penetration ability and cell toxicity and aSOs of longer constructs with higher membrane disruption power and cell toxicity. Only NTD and NAC regions are known to interact directly with vesicles in monomeric aSN[61,62], but truncation of CTD may have indirect effects by altering the core-shell distribution within the oligomer. A structural study on αSO formation and membrane penetration showed a division of labor in which the NTD provides interfacial contacts while the β-sheet-rich core penetrates into the membrane[56]. Even though CTD truncation provides greater freedom for the NTD, a thicker shell covering the β-sheet core (a consequence of CTD truncation) may restrict the β-sheet core's membrane penetration and thus reduce cell toxicity.

In summary, C-terminal truncation of αSN results in a very sudden increase in the rate of fibril replication upon truncation beyond residue 121 due to an increase in the secondary processes of fibrillation., most likely fragmentation This effect is apparent even when only 10% of the aggregating protein present is the truncated variant. By contrast, truncation beyond residue 121 essentially abolishes oligomerization. Taken together, our results show that though outside the actual aggregate core, residues in the region of residue 121 onwards act as an electrostatic double-edged safety guard, supressing fibrillation while being prerequisites for oligomerization. Several CTD truncated αSNs have been found in neural cells or Lewy bodies[15,63]. Our work identifies a potential "danger zone" of particularly deleterious CTD truncations, namely a CTD truncation range from αSN 121 to αSN 125 which shows increased fibrillation compared to αSNFL while retaining the ability to form cytotoxic αSN oligomers on their own. In this context it is striking that the most common in vivo truncated forms of αSN are αSN119 and αSN122[11]. This highlights the deleterious consequences of biological processing of αSN.

## Methods

**Protein purification and sample preparation**. All proteins were expressed in *E. coli* BL21 (DE3) cells. αSN FL (human wild-type αSN) was expressed with pE11-D plasmid while all αSN mutants were expressed with a pET15-B plasmid with a stop codon before the His-tag. All plasmids were transformed by electroporation and expressed using autoinduction media as described[64]. Constructs αSN121 and longer (whose net charge is negative at pH 7.4) were purified on an anion exchange column (HiTrap™ Q HP) while constructs αSN115 and shorter (whose net charge is positive at pH 7.4) were purified on a cation exchange column (HiTrap™ SP HP). Purified protein was dialyzed in MQ water and lyophilized. Just prior to use, the lyophilized protein was dissolved in PBS (13 mM phosphate, 137 mM NaCl, 3 mM KCl, pH 7.4) and filtered through a 0.2 μm nitrocellulose membrane filter to remove any pre-existing aggregates. Protein concentration was measured on a Nanodrop 1000 spectrophotometer (Thermo Scientific) using theoretical extinction coefficients at 280 nm based on the content of Tyr residues (between 1 and 4 depending on construct).

**ThT fibrillation assays**. Fibrillation kinetics was monitored by Thioflavin-T (ThT) fluorescent assay in black polystyrene 96-well plate using a TECAN Infinite M200 microplate reader pre-adjusted to 37 °C. 3-mm glass beads were used for experiments with shaking (300 rpm, 10 min shaking in 12 min measurement intervals). Shaking for low-seed fibrillation assay was done only 20 s in every 10 min measurement intervals. Freshly sonicated fibrils (probe sonicated for 15 s) were used for seeding experiment (performed in quiescent mode). 40 μM ThT was used for all experiments.

### Analysis of fibrillation kinetics

*Description of model*. Normalized ThT fibrillation data were fitted to a molecular mechanism of filamentous aggregation that includes primary nucleation, fragmentation, and saturating elongation. The differential equations describing the time evolution of aggregate number concentration, P(t), and aggregate mass concentration, M(t), are

$$\frac{dP}{dt} = k_n m(t)^{n_c} + k_- M(t) \tag{1}$$

$$\frac{dM}{dt} = 2k_+ \frac{m(t)}{\left(1 + \frac{m(t)}{K_E}\right)} P(t) \tag{2}$$

where $k_n$, $k_-$ and $k_+$ are the rate constants of primary nucleation, fragmentation and elongation, $K_E$ is the saturation concentration of elongation, $m(t)$ is the monomer concentration at time $t$ and $n_c$ is the reaction order of primary nucleation. Approximate analytical solutions to these equations, the integrated rate laws[39], are used to fit the data. Note that if secondary nucleation instead of fragmentation was the dominant process by which new fibrils are formed, $k_-$ in the above equation would be replaced by $k_2 m(t)^{n2}$, where $k_2$ and $n_2$ are the rate constant and reaction order of secondary nucleation. When $n_2 = 0$, the two expressions are mathematically equivalent, making it impossible to distinguish fragmentation from saturated secondary nucleation based on fits of $M(t)$ and $P(t)$ alone[40]. A detailed investigation to distinguish fragmentation from secondary nucleation in αSN aggregation at low pH can be found in Gaspar et al.[41]

**Effective elongation rates**. These were obtained using ThT fibrillation assays with 50 μM of αSN monomer and 2.5–12.5 μM seeds (monomer units), corresponding to 5–25% of monomer. To calculated the effective rate of elongation[65], initial rates of fibrillation were plotted against the seed concentration and the slope of the fitted line was used to determine the effective elongation rate. The initial rate of increase in the presence of significant amounts of seeds is approximated by:

$$\frac{dM}{dt} = 2k_+ \frac{m_0}{\left(1 + \frac{m_0}{K_E}\right)} P_0 \tag{3}$$

where $m_0$ is the initial monomer concentration and $P_0$ is the number concentration of seed fibrils. Both $m_0$ and $K_E$ are known, the latter from the fits of the unseeded data as described above, so we can determine the product $P_0 k_+$. However, the initial seed number concentration, $P_0$, (not to be confused with the seed mass concentration $M$) is difficult to determine accurately. We are here interested in the relative variations of the rates as the degree of truncation changes, so rather than estimate $P_0$ to obtain absolute values for $k_+$, we instead compare the relative values of $P_0 k_+$, setting the value for the full-length protein to 1. Under the assumption that the length of the sonicated seeds does not vary with truncation, relative changes in $P_0 k_+$ are equivalent to relative changes in the elongation rate constant $k_+$.

**Oligomer preparation**. To increase yields of αSN oligomers, each αSN construct (initially lyophilized after purification as described above) was subjected to two cycles of dissolution to ~10 mg/ml in milliQ followed by freeze-drying. Subsequently, the construct was dissolved to 8 mg/mL in PBS and incubated at 37 °C for 4 h in 1.5 microtubes while shaking at 900 rpm. Large aggregates were spun down at 12,000 × g for 5 min. The supernatant was applied on a SEC column (Superose 6 Prep Grade column; GE Healthcare Life Sciences, Sweden) by AKTA system (GE Healthcare) and oligomer fractions were collected for further analysis. 100-kDa centrifuge filters (Amicon®) were used to concentrate the collected oligomers. To prepare oligomers in different pH and salt concentrations, monomers were first dissolved in milliQ water and then diluted 1:1 to the desired working concentration with 2× concentrated buffers.

**DMPG vesicle preparation**. DMPG vesicles were prepared as described[66]. Briefly, 2 mg of 1,2-dimyristoyl-sn-3-phosphatidylglycerol (DMPG) was suspended in 1 mL PBS and freeze-thawed in liquid nitrogen and a 55 °C water bath respectively, 10 times. The solution was extruded 21 times through a 200 nm filter (Whatman, Nuclepore TE membrane). To prepare calcein-filled vesicles, 10 mg DMPG was suspended in 1 mL PBS including 70 mM calcein with the same procedure as normal vesicle preparation. Then, the excess calcein was removed by PD-10 desalting column (GE Healthcare) and the fractions containing calcein were identified by monitoring fluorescence (VarioScan Flash fluorimeter (Thermofisher Scientific); excitation at 495 and emission at 515 nm with a 5 nm bandwidth) after adding Triton X-100. Vesicles were prepared and used within 1 day.

**CD spectroscopy**. Far-UV CD spectra were recorded on a Jasco J-810 spectrophotometer (Jasco Spectroscopic Co. Ltd., Japan) from 250 to 195 nm using 0.2 mg/ml of αSN monomers or oligomers in a 1 mm cuvette.

**Small-angle X-ray scattering (SAXS)**. SAXS was carried out on an optimized NanoSTARSAXS instrument[67] from Bruker AXS at Aarhus University equipped

with scatterless slits[68] and a liquid Ga metal jet X-ray source from Excillum[69], which measures at a wavelength of $\lambda = 1.34$ Å. The SUPERSAXS program package (Oliveira, C.L.P. and Pedersen J.S., un-published) was used for background subtraction and to convert data to absolute scale. Water was used as a calibration standard and the data is plotted as a function of the scattering vector $q = 4\sin(\theta)/\lambda$. SEC-purified oligomers were measured at a concentration ranging from 0.19 to 1.35 mg/mL depending on the yield and were measured for 30 min at 20 °C. Model independent information of the oligomers was obtained by the Indirect Fourier Transformation (IFT) procedure[70] that gives the pair distance distribution function $p(r)$ that gives real-space distance information. To obtain further insight in the oligomer structure, we modeled the data with a previously developed aSN oligomer model[66] where the oligomer is assumed to have a prolate ellipsoid shape with a compact core and a shell of flexible polymer-like protein. By modeling the data on absolute scale, the average number of monomers in each complex ($N_{chain}$) can be obtained. Furthermore, the long ($r_{long}$) and short ($r_{short}$) axis of the core, the thickness of the shell ($D_{shell}$) and the number of amino acids in in the shell can be modeled.

**Transmission electron microscopy (TEM)**. 5 μL of 0.2 mg/mL oligomer or 0.4 mg/mL fibril solution was applied to glow-discharged 400-mesh carbon-coated copper grids for 60 s before being removed with filter paper and immediately stained with 3 μL of a 2% (w/v) aqueous uranyl formate solution for 15 s. The grid was stained again for 15 s before excess stain was removed by blotting with filter paper. TEM was performed on a Tecnai™ G2 Spirit transmission electron microscope operated at 120 kV. Digital acquisitions were performed with a bottom-mounted TVIPS camera.

**Calcein release assay**. Calcein release was performed in triplicates and in the range of 0.001 to 100 μM of monomers and 0.00005 to 10 μM of oligomers as described[66]. Briefly, 2 μL of vesicle was added to 138 μL PBS in a 96-well plate and then 10 μL of protein solution was added to each well. The fluorescence of calcein (excitation at 495 and emission at 515 nm) was monitored by VarioScan Flash fluorimeter (Thermofisher Scientific) using a 5 nm bandwidth. After 1 h, 2 μL of Triton X-100 was added to each well to lyse all vesicles and release all calcein. Calcein release (CR) was calculated according to:

$$CR\% = 100 \times (F_p - F_c)/(F_f - F_c) \tag{4}$$

where $F_p$ and $F_c$ are the fluorescence intensities after 1 h incubation with αSN and PBS only, respectively, (either monomer or oligomer) and $F_f$ is the fluorescence intensity after addition of Triton X-100. The data were plotted in KaleidaGraph software (V 4.5) and fitted with a sigmoidal model. CR$^{50}$% is the concentration of protein required to disrupt 50% of vesicles.

**Cell toxicity assay**. MTT (3-[4,5-dimethylthiazol-2-yl] -2,5-diphenyltetrazolium bromide) assay was used to measure the cell viability after 24 h treatment with different variants of C-terminally truncated αSN oligomers. SH-SY5Y cells were seeded in 96-well plates at a density of $5 \times 10^4$ cell/ml in DMEM media supplemented with 10% FBS, 100 units/ml penicillin and 100 μg/ml streptomycin. The cells were incubated in a humidified atmosphere incubator with 5% $CO_2$ and 95% humidity at 37 °C for 24 h. The culture medium was then replaced with fresh medium containing 10 μM α-SN oligomers and incubated for another 24 h. Then the cell medium was replaced with fresh medium containing 20% MTT (5 mg/ml) and the cells were incubated for an additional 4 h at 37 °C. Subsequently 100 μl DMSO was added to dissolve formazan crystals by incubating for 30 min on a shaker at room temperature. Absorbance was measured on a plate reader at 570 nm.

**Statistics and Reproducibility**. Every experiment generally tests different concentrations with three repeats. In Fig. 1, every data points correspond to a repeat. In Fig. 2a–c, every data point corresponds to a separate experiment with a series of concentrations, while in Fig. 2e–g, curves are the average of two repeats. SAXS experiment was done once for each oligomer. Calcein release experiment has been done with three repeats and cell toxicity with 4 repeats. All the error bars show standard deviation except for Fig. 4h which is 95% confidence interval (CI). One-way ANOVA was performed for comparisons in Fig. 4h using GraphPad prism version 8 (GraphPad Software, San Diego, California USA).

**Reporting summary**. Further information on research design is available in the Nature Research Reporting Summary linked to this article.

## Data availability

Source data for graphs and charts in the main figures is provided as Supplementary Data 1 and any remaining information can be obtained from the corresponding author upon reasonable request. Supplementary Data 1.xlsx file include 8 data sheets named after main figures and panels.

## Code availability

No custom code was used in this work.

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

## Acknowledgements
D.E.O. (grant R276-2018-671) and A.K.S (grant R287-2018-1836) are grateful to the Lundbeck Foundation for generous and sustained support. We also thank the Novo Nordisk Foundation for supporting D.E.O. and P.A. (grant NNF17OC0028806). We thank Dr. Frans Mulder for inspiring discussions about αSN electrostatics.

## Author contributions
A.F. and D.E.O. conceived the project. A.F. carried out all protein purification and aggregation studies, performed all kinetic analysis and wrote the first draft. D.E.O. supervised all aspects of the project, obtained funding, carried out project administration and edited the manuscript. J.N.P. and J.S.P. carried out SAXS experiments and analysis. G.M. and T.K. helped analyse kinetic data. A.K.S. carried out data analysis. P.A. measured cell viability. L.G. provided truncated αSN constructs, supervised by M.A.N. and A.F.S.

## Competing interests
The authors declare no competing interests.
