## [Peer Review File · Communications Biology]

Reviewers' comments:

Reviewer #1 (Remarks to the Author):

The authors reveal the role of the C-terminal tail of α SN in the molecular mechanism of fibrillation and the formation and oligomeric species. Using fibrillation assay shows that removal of more than 15 residues strongly enhances fibrillation thanks to an increased rate of replication. By applying SAXS, they visualized α SN oligomers in solution, and they also show that α SN CTD truncations abolished oligomerization, and only more extensive, less stable oligomers are formed. Using calcein assays, they show that CTD truncations mutant are less disruptive towards lipid vesicles. These biophysical characterizations were linked with in-vitro studies using cell toxicity assays. In summary, the article is very well written, easy to read, and complements the subject's bibliography well. However, although the authors had a rigorous approach with the SAXS, fibrillation assay, calcein assays, and in vivo assays, I have some questions and remarks in my report.

Minor comments :

1) Please comment on the small primary nucleation rates for α SN115 shown in Fig. 2B).

2) Page 6 : (shown in Fig. 2C) should be (Fig. 2C).

3) Please explain the meaning of [monomer]. Is there a reason not to use monomer instead?

4) Page 7: Authors claim that secondary fibrillation is most likely controlled by "The fact that deficient concentrations of seeds (0.03%) strongly promoted the fibrillation demonstrates the dominance of secondary processes which is most likely fragmentation."
Could the preparation of seeds by sonication lead to oxidation of α SN that may impact the fibrillation kinetics rather than assume the simple fragmentations?

5) Page 7: please add the reference for in vivo studies showing truncated α SN accelerate fibrillation/aggregation in vivo. "...fast aggregating truncated versions, even when the full-length protein is the predominant form and truncated versions are present only in small proportions, as may be the case in vivo."

6) Page 8: For better readability, please add a cartoon description of the core-shell model in Fig 3. Or Figure S6.

7) Page 8: I'm aware of previous manuscripts describing the α SN -core-shell model by SAXS and HDX. However, please add the reference and maybe a short description of methods for determining AA in shell and # of monomers in oligomer related to Fig 3C and 3D.

8) Fig 3: please separate the SAXS Fig3 A-D panels (new figure) from the determination of the ability to permeabilize negatively charged vesicles Fig 3 E-J

9) Discussion page 11: Only for your consideration: it will be valuable to interpret oligomer model with coarse grain model or pseudo atomistic model. It brings value to the study if authors directly compare SAXS (solution state) of α SN and α SN-ANO with $P(r)$, Volumes, or models. Representation of SAXS data with more detailed analysis of $P(r)$ functions or models will avoid the comments like "similar shape, "similar modes of action in promoting interactions between different parts of α SN, "shorter constructs are more elongated with relatively bigger shells."

10) Please check the meaning of α SO > α SN

Reviewer #2 (Remarks to the Author):

The authors present a comprehensive analysis of a range of C-terminal truncations of α SN, linking the importance of high C-terminus charge for decreased fibrillation rates. Removal of charge increases aggregation rates, including when incubated with FL α SN. The ability to form oligomers, to disrupt synthetic vesicles and cell toxicity was reduced with truncated α SN. Some of

the materials don't have a very strong physiological link, e.g. oligomers formed by freeze drying, synthetic lipid membranes and toxicity studies on SHSY5Y cells which are very different from dopaminergic neurons. However, the authors provide a robust analysis of the data they have provided which will aid understanding of the intramolecular interactions of aSN which promote/inhibit aggregation.

Major comment:

1) I strongly believe all ThT-based kinetic assays should be accompanied with data to show whether the ThT fluorescence signal is reflective of aggregation rate of the different species. There are a few papers now showing different ThT fluorescence intensities from different polymorphs of amyloids. The authors should provide some additional data to show that the ThT fluorescence reflects the fibril formation. For instance, quantification of the remaining monomer and/or of the formed fibrils can be done by quantification of protein concentration of either species, reversed phase chromatography, gel filtration, stained SDS-PAGE gels, performed at the end of the assays.

Minor comments:

- 2) Figure 1 – what does each point on the graph represent?
- 3) Figure 1 - Is aSN121 showing a single linear correlation? This graph looks more similar to aSN110 than the longer constructs.
- 4) Page 7, line 178 – are oligomers formed by rounds of freeze-drying physiologically relevant?
- 5) Page 7, line 181 – where in figure S5DE do the authors have the NaCl data? How does this differ to the PBS buffer (that I assume the other oligomers are in?). Buffer differences have been shown to be important for fibril polymorph formation, this could be interesting for oligomers too.
- 6) Page 8, line 186 – The peak size for the aSOs in Figure S5G don't look to give the same yield for 121 and 125 compared to 130 and 135.
- 7) Page 8, line 188-189 – It's not clear why ONE was used or how this is aiding oligomer formation of shorter fragments or why this is useful to investigate.
- 8) Page 9, line 216 – Is there a similar mechanism linking calcein release and cell toxicity? This seems a bit of a weak or presumptive link.
- 9) Discussion – What are the most abundant truncated species identified in brain tissue? Are there species identified in this work that promote fibrillisation of FL aSN, cell toxicity and calcein release that have also been identified in vivo and may thus be more important to investigate further?
- 10) Page 12, line 281-294 – This section is not very clear, I'm not sure what the use of the ONE oligomers is showing, that you need N and C terminal interactions to form oligomers?
- 11) Page 12, line 295-303 – Are changes in the charge of the protein or aSOs upon truncation likely to affect the interaction of aSN with the vesicles?
- 12) Supplementary Page 1, line 24 – What ThT concentration was used?
- 13) Supplementary Page 7 and 8 - Figure S2 and S3 could the authors please provide the number of replicates and wells used in the data.
- 14) Supplementary Page 10 - Figure S5a-e – are these FL aSN?

Reviewer #3 (Remarks to the Author):

In the paper entitled 'The C-terminal tail of α -synuclein protects against aggregate replication but is critical for oligomerization', the authors describe the results of a study aimed at determining effects of C-terminal truncations on the self-assembly properties of alpha-synuclein (aSyn), a protein that forms aggregates in the brains of individuals with Parkinson's disease (PD) and other synucleinopathy disorders. The investigators report that aSyn truncation variants with 121 or fewer residues have a substantially increased propensity to form amyloid-like fibrils, apparently because they more readily undergo secondary fibrillization as a result of enhanced fragmentation. An increased rate of fibrillization is also observed in mixtures of full-length aSyn with as little as 10% (mol/mol) of the aSyn-110 truncation variant. In contrast, C-terminal truncation interferes with aSyn oligomerization, and oligomers formed by truncation variants in the presence of the pro-oxidant 4-oxo-2-nonenal have a reduced propensity to permeabilize membranes and elicit cytotoxicity. The authors attribute these effects of the aSyn C-terminal tail to its electrostatic interactions with the N-terminal and NAC (central hydrophobic) domains, which favor oligomerization but interfere with the formation of amyloid-like fibrils.

This paper is significant in terms of its relevance to molecular mechanisms involved in the pathogenesis of PD and other synucleinopathy disorders. In particular, the results are of a high impact because they yield important insights into structural features of the C-terminal domain of aSyn that modulate two key pathogenic events – the protein's conversion to potentially neurotoxic oligomers and the formation of aSyn amyloid-like fibrils that can spread throughout the brain. The findings reported here are novel and represent an important advance in the aSyn field.

Strengths of the paper in addition to its significance include the fact that it is well written and organized, and the authors have been thorough in terms of referencing the key literature of relevance to this study. The experiments have been carefully executed, yielding high-quality results, and the data processing and mathematical modeling are rigorous. On the other hand, there are several errors in the text and figures/figure legends, and parts of the paper are unclear. In addition, some of the authors' conclusions would be strengthened with additional discussion or by including additional data. These issues should be addressed in a revised version of the manuscript.

Specific comments:

p. 4, line 86, 'is still lacking'; delete

p. 5, line 95, 'the CTD (residues 101-140)'; previously the CTD was described as a segment spanning residues 96-140 (the same polypeptide segment should be referred to throughout the paper).

p. 5, line 110, 'high concentrations with a float slope'; replace 'float' with 'flat'.

p. 6, lines 123-125, 'and the rate-limiting step shifts from monomer binding to the end of fibrils (which is dependent on [monomer]) to the conformational conversion of the monomer to the fibrillated state (which is not)'; it is unclear why the [monomer]-independent phase at high monomer concentrations is interpreted to reflect a conformational change of monomer to the fibrillated state (a component of the primary fibril elongation phase), as opposed to the secondary [monomer]-independent processes of 'fragmentation or saturated secondary nucleation' referred to earlier (lines 117-118). It would be helpful if this point were clarified.

p. 6, lines 138-139, 'from plots of the initial rate of fibrillation against the total concentration of monomer and seed'; the plot in Fig. S4 shows only the seed concentration, not the total concentration of monomer and seed, on the x-axis (this point should be clarified).

p. 6, lines 142-143, 'To compare different constructs with different reaction orders of nucleation, we calculated the nucleation rate at 50 micromolar'; it would be helpful if a justification were provided for choosing 50 micromolar as the optimal concentration to account for different reaction orders.

p. 6, line 153, 'is around -0.5 for all mutants'; the value is closer to -0.3 to -0.4.

p. 7, line 158, 'though an increase in the turnover rate on the surface of fibrils'; it would be helpful if this concept were explained in greater detail (turnover of what?)

p. 7, lines 162-163, 'demonstrates the dominance of secondary processes which is most likely fragmentation'; additional detail should be provided to explain why the authors conclude that the stimulatory effect of agitation on the fibrillization rate reflects enhanced fibril fragmentation rather than turnover.

p. 7, line 169, '60% aSN110 and higher showed the same behavior as 100% aSN110'; delete 'and higher' because 60% was the highest percentage of aSyn 110 tested in this experiment.

p. 7, line 180, 'without any change in structure according to Small Angle X-ray Scattering'; refer to

Fig. S5C here.

p. 8, line 206, 'and both aSOs and ONE-modified aSOs had diameters of approx. 10 nm'; it is unclear why the larger sizes of aSOs prepared from shorter C-terminal aSyn variants (determined via SAXS analysis) wasn't evident in the TEM images. This point should be addressed in the discussion.

p. 8, lines 213-214, 'ONE-modified aSOs of shorter constructs (ONE-aSO100 to ONE-aSO125) showed reduced disruption (Fig. 3IJ)'; replace 'ONE-aSO125' with 'ONE-aSO121'.

pp. 8-9, lines 214-215, 'SH-CY5Y neuroblastoma cells'; replace 'SCY5Y' with 'SH-SY5Y'.

p. 10, lines 225-226, 'Although elongation of CTD truncated aSNs was slowed down 4-fold in the shortest construct at 50 micromolar aSN due to saturation of elongation'; it would be helpful if the data supporting the '4-fold' value in this statement were referred to here, as this is not evident in the data in Figs. 2A and S4.

p. 10, line 240: reference 59 is missing from the bibliography.

p. 10, lines 248-251, 'mutants containing disulfide bonds between residues 107 and 124 in CTD combined with disulfide bonds in the NTD-NAC region (residues 9, 42, 69 and 89) do not fibrillate; by contrast disulfide bonds in the NTD-NAC region alone are insufficient to inhibit fibrillation'; this statement should be reworded to indicate that intramolecular disulfide bonds linking cysteine residues introduced in the CTD and NTD-NAC regions were used in the study described in reference 51.

Fig. 2A-C: it should be explained why there are replicate points for some of the aSyn truncation variants (aSyn 105 in panels A and C; aSyn 125 and aSyn 130 in panels B and C; aSyn 135 in panel C).

Fig. 2E-F: it should be made clear in the figure legend and in the corresponding part of the Results section that these experiments were carried out with full-length aSyn. Also, the paper would be strengthened by demonstrating that similar results are obtained when repeating these experiments with at least a subset of truncation variants (e.g. aSyn 125 and aSyn 130, to account for the pronounced difference in the secondary rate constants determined for these two variants in terms of a difference in the degree of fragmentation or secondary nucleation).

Fig. 2G: it should be indicated on the graph that the ratios in the legend refer to 'aSyn FL: aSyn 110'.

Figs. 3, S6, S7 legends: the number of replicate data points and the method used to assess statistical significance should be indicated.

Fig. 3 and Fig. S6H legends: the error bars should be defined (e.g. SD or SEM).

Fig. 3D legend: specify that the number of monomers per oligomer for both kinds of oligomers increases with the extent of truncation.

Fig. S2: it would be helpful to show the curve fits for aSyn 135, aSyn 130, and aSyn 125 for the secondary nucleation dominant and fragmentation dominant models separately, and to include MRE values on the plots as was done for aSyn WT. Also, the following statement in the legend, 'the models with saturation fit just as well', should be clarified (e.g. refer to 'the secondary nucleation dominant and fragmentation dominant models with saturation elongation, which produce a better fit with aSyn variants with more extensive C-terminal truncations (Fig. S3)').

Fig. S5C: the caption in the graph is difficult to read and should be enlarged.

Fig. S5D, legend: the meaning of the asterisks written beside some of the pH values should be explained.

Fig. S5F, legend: 'between pH 6.5 and 9.5' should be written as 'between pH 6 and 9.5'.

Fig. S5G, legend, 'aSN 115 and shorter constructs'; delete 'and shorter constructs' because aSyn 115 was the shortest variant examined here.

Fig. S6B: a single label at the top of the graph should be used to indicate that the curves correspond to a set of ONE-treated aSyn oligomers, rather than repeatedly inserting 'ONE' in all of the curve labels below (currently, the curve labels are difficult to see because of the crowded nature of the graph).

Fig. S6D,F: the curve labels are difficult to see because of the crowded nature of the graph, and the term 'ONE' should be used at the top of the graph rather than repeatedly in the individual curve labels.

Fig. S6G: correct the typo in the x-axis label.

General comment about the Methods' section: an additional subsection describing the methods used for statistical analysis should be included.

Supplement, p. 2, line 45, 'Gaspar et al. [40]'; the Gaspar et al. reference is listed as reference 41 in the bibliography.

Supplement, p. 3, line 61, 'Oligomer Preparation'; additional detail explaining how the experiments represented by Fig. S5D-F should be provided.

Supplement, p. 3, lines 83-87, 'Fibrils were sonicated to minimize light scattering and spectra were recorded from 250 to 185 nm in 1 mg/ml using 0.1 mm cuvette. All data are averages of three repeats with 0.5 nm steps per 0.5 s and 1 nm bandwidth. Thermal scans to monitor conformational changes upon vesicle binding were carried out as described. Briefly, a solution of 0.2 mg/mL monomeric aSN and 1 mg/mL DMPG was heated from 5 to 95 deg C at 1 deg C/min and ellipticity was monitored at 220 nm'; this text could be deleted because CD data are only shown for oligomeric aSyn in the absence of phospholipids (Fig. S6E,F).

Supplement, p. 4, line 105, 'or 0.4 mg/mL fibril'; delete.

Supplement, p. 5, line 125, 'different variants of N- terminal truncated'; this should read 'C-terminal truncated'.

Supplement, p. 5, line 127: presumably the streptomycin concentration was 100 micrograms per mL, not 100 mg/mL?

Reviewer #1 (Remarks to the Author):

The authors reveal the role of the C-terminal tail of α SN in the molecular mechanism of fibrillation and the formation and oligomeric species. Using fibrillation assay shows that removal of more than 15 residues strongly enhances fibrillation thanks to an increased rate of replication. By applying SAXS, they visualized α SN oligomers in solution, and they also show that α SN CTD truncations abolished oligomerization, and only more extensive, less stable oligomers are formed. Using calcein assays, they show that CTD truncations mutant are less disruptive towards lipid vesicles. These biophysical characterizations were linked with in-vitro studies using cell toxicity assays. In summary, the article is very well written, easy to read, and complements the subject's bibliography well.

Response: We thank the reviewer for this generous assessment and for the constructive suggestions for improvement which we address below.

However, although the authors had a rigorous approach with the SAXS, fibrillation assay, calcein assays, and in vivo assays, I have some questions and remarks in my report.

Minor comments:

- 1) Please comment on the small primary nucleation rates for α SN115 shown in Fig. 2B.

Response: We appreciate the reviewer's keen observation. The values for α SN115 do stand out. Analysis is affected by the fact that the rates of primary and secondary nucleation are coupled to some extent in the fits, i.e. a decrease in primary rate can be compensated to some extent by an increase in secondary rate. This effect is apparent for SN115 to some extent. In addition, primary nucleation, due to the potential catalytic effects of plate surfaces and the air water interface, is more prone to experimental variation. Given these observations, we do not believe that the low primary rate of SN115 is indicative of a fundamentally different behaviour of SN115 to SN110 and SN121 but acknowledge the need to point this out. Accordingly, we note in the text now that "(we attribute the low k_n value of α SN115 to a fitting artifact in which a low k_n value is compensated by a higher k_c value)."

- 2) Page 6: (shown in Fig. 2C) should be (Fig. 2C).

Response: Duly corrected.

- 3) Please explain the meaning of [monomer]. Is there a reason not to use monomer instead?

Response: We use this term for text economy to indicate the concentration of monomer. To make it clear we explain this the first time we use it.

4) Page 7: Authors claim that secondary fibrillation is most likely controlled by “The fact that deficient concentrations of seeds (0.03%) strongly promoted the fibrillation demonstrates the dominance of secondary processes which is most likely fragmentation.”

Could the preparation of seeds by sonication lead to oxidation of α SN that may impact the fibrillation kinetics rather than assume the simple fragmentations?

Response: We would like to clarify that we use sonicated fibrils both for shaking and quiescent conditions (Fig. 2E); thus we carry out a comparative study using fibrils that have been sonicated to the same extent but are left to elongate under different conditions. However, it is a thought-provoking suggestion that sonication could lead to fibril oxidation and thus compromise seeding quality. We are not aware of any such studies in the literature and have not seen any evidence for this in our own studies with e.g. MALDI-TOF MS (though we must admit that we have not looked specifically for these modifications). We suspect that it will be difficult to gauge these effects as fibrils are typically protected against chemical insults better than monomers and the effects of oxidation may be difficult to discern within the usual errors. We are in fact embarking on a separate study to study the effect of sonication on fibril size distribution (using functional bacterial amyloid) and will bear this in mind in our analysis. Thank you for raising this!

5)Page 7: please add the reference for in vivo studies showing truncated α SN accelerate fibrillation/aggregation in vivo. “...fast aggregating truncated versions, even when the full-length protein is the predominant form and truncated versions are present only in small proportions, as may be the case in vivo.”

Response: We now provide the following two references:

- Li, W. *et al.* Aggregation promoting C-terminal truncation of α -synuclein is a normal cellular process and is enhanced by the familial Parkinson's disease-linked mutations. *Proceedings of the National Academy of Sciences of the United States of America* **102**, 2162-2167, doi:10.1073/pnas.0406976102 (2005).
- Ulusoy, A., Febbraro, F., Jensen, P. H., Kirik, D. & Romero-Ramos, M. Co-expression of C-terminal truncated alpha-synuclein enhances full-length alpha-synuclein-induced pathology. *European Journal of Neuroscience* **32**, 409-422, doi:10.1111/j.1460-9568.2010.07284.x (2010).

6) Page 8: For better readability, please add a carton description of the core-shell model in Fig 3. Or Figure S6.

Response: Great idea. We added a sketch of the α SN FL oligomer with indicated dimensions to Fig. 3 with the legend “(E) Schematic model of unmodified α SN FL oligomers based on SAXS ¹. A compact core is surrounded by a shell of more flexible polypeptide chains.

7)Page 8: I’m aware of previous manuscripts describing the α SN -core-shell model by SAXS and HDX. However, please add the reference and maybe a short description of methods for determining AA in shell and # of monomers in oligomer related to Fig 3C and 3D.

Response: The reviewer is quite right. We now provide the reference (Lorenzen, N. et al. The role of stable α -synuclein oligomers in the molecular events underlying amyloid formation. Journal of the American Chemical Society 136, 3859-3868, doi:10.1021/ja411577t (2014)). In the supplementary information under SAXS we write that “we modelled the data with a previously developed α SN oligomer model⁶ where the oligomer is assumed to have a prolate ellipsoid shape with a compact core and a shell of flexible polymer-like protein. By modelling the data on absolute scale, the average number of monomers in each complex (N_{chain}) can be obtained. Furthermore, the long (r_{long}) and short (r_{short}) axis of the core, the thickness of the shell (D_{shell}) and the number of amino acids in in the shell can be modelled.” We hope this is sufficient information. Reference 6 (Lorenzen et al JACS) provides detailed information about how to distinguish between a compact core and a flexible shell.

8) Fig 3: please separate the SAXS Fig3 A-D panels (new figure) from the determination of the ability to permeabilize negatively charged vesicles Fig 3 E-J

Response: Good idea. In the revised manuscript we have a figure 3 with old panel A-D together with the schematic view of an α SN FL oligomer (panel E, cfr. points 6-7) and a figure 4 with calcein release and cell toxicity.

9) Discussion page 11: Only for your consideration: it will be valuable to interpret oligomer model with coarse grain model or pseudo atomistic model. It brings value to the study if authors directly compare SAXS (solution state) of α SN and α SN-ANO with P(r), Volumes, or models.

Representation of SAXS data with more detailed analysis of P(r) functions or models will avoid the comments like “similar shape, “similar modes of action in promoting interactions between different parts of α SN, “shorter constructs are more elongated with relatively bigger shells.”

Response: We appreciate the reviewer’s thoughts on this and sympathize with the desire to provide more structural details. Unfortunately we find it difficult to advance further in our current analysis of the modelling. Our current models agree completely with our SAXS data and would therefore also be entirely consistent with p(r) functions from indirect Fourier transformation. Furthermore, please note that volumes of models are actually difficult to define as the shell of the oligomer has a

lower density of protein than the compact core. We and others have tried to make *ab-initio* bead modelling of the SAXS data without any assumptions on the structure (cfr. Giehm et al. PNAS 2011) which leads to a similar structure as our model, namely an ellipsoidal shape. However, such modelling is done on a somewhat incorrect basis, since the SAXS intensity contains significant scattering from the random coil chains; this cannot be included properly as it requires an ensemble of particles to give the correct q dependence - and this is not possible with the methods currently available. The only way around this is to omit the high- q part of the SAXS intensities and then the models are of even lower resolution than usual for SAXS and cannot reproduce a core-shell structure. In a nutshell, these methods assume compact globular structures and therefore cannot really investigate a core-shell structure. We therefore believe it is better to restrict ourselves to the current level of resolution.

10) Please check the meaning of $\alpha_{SO} > \alpha_{SN}$

Response: Unfortunately we are not sure what the reviewer means by this. We do not use any such expression in our document. We will be happy to clarify if the reviewer can expand on this.

Reviewer #2 (Remarks to the Author):

The authors present a comprehensive analysis of a range of C-terminal truncations of α SN, linking the importance of high C-terminus charge for decreased fibrillation rates. Removal of charge increases aggregation rates, including when incubated with FL α SN. The ability to form oligomers, to disrupt synthetic vesicles and cell toxicity was reduced with truncated α SN. Some of the materials don't have a very strong physiological link, e.g. oligomers formed by freeze drying, synthetic lipid membranes and toxicity studies on SHSY5Y cells which are very different from dopaminergic neurons. However, the authors provide a robust analysis of the data they have provided which will aid understanding of the intramolecular interactions of α SN which promote/inhibit aggregation.

Response: We thank the reviewer for this fair and constructive assessment.

Major comment:

1) I strongly believe all ThT-based kinetic assays should be accompanied with data to show whether the ThT fluorescence signal is reflective of aggregation rate of the different species. There are a few papers now showing different ThT fluorescence intensities from different polymorphs of amyloids. The authors should provide some additional data to show that the ThT fluorescence reflects the fibril formation. For instance, quantification of the remaining monomer and/or of the formed fibrils can be done by quantification of protein concentration of either species, reversed phase chromatography, gel filtration, stained SDS-PAGE gels, performed at the end of the assays.

Response: We enthusiastically concur with the reviewer's healthy skepticism about relying blindly on ThT fluorescence. Specifically, we determine the extent of fibrillation by measuring the remaining monomers at the end of reactions. We measured the end stage monomer concentration by running an SDS-PAGE for each mutant at the end of reaction after spinning down the aggregates. We checked the leftover monomer after fibrillation reach a plateau for all mutants at 70 μ M concentration and now provide two new figures (Fig. S4 and Fig. S5). Fig. S4 shows that 66 to 85% of monomer are incorporated into fibrils for α SN FL to α SN 125 while almost the entire α SN population has been fibrillated for shorter mutants. Amylofit analysis involves normalizing the ThT fluorescence levels and therefore does not rely on absolute fluorescence values. Normalized data are perfectly adequate for Amylofit analysis provided most of the protein is aggregated. Thus we confirm that the ThT fluorescence time profiles actually reflects fibril formation and that the fibrils are similar in shape to full-length α SN (Fig. S5). To reflect this response, we now add that "SDS-PAGE analyses confirmed that the great majority of α SN had aggregated by the end of the reaction

(**Fig. S4**). (Note that slightly incomplete levels of fibrillation will not affect the outcome of the analysis in Amylofit). TEM demonstrated very similar architecture for the fibrils formed by the different truncation variants (**Fig. S5**).”

Minor comments:

2) Figure 1 – what does each point on the graph represent?

Response: This explanation is now added to the legend of the graph: “Each point represents the half-time of a ThT curve (*i.e.* the time to reach half the maximal fluorescence value) from a fibrillation experiment carried out at the indicated α SN concentration.”

3) Figure 1 - Is aSN121 showing a single linear correlation? This graph looks more similar to aSN110 than the longer constructs.

Response: The reviewer is quite right. As we mention in the text, aSN121 is the first mutant (starting from α SNFL) that starts to deviate from its longer counterparts. This can be seen in double logarithmic plot with a broken linear correlation. Although α SN125 looks similar to shorter mutants in terms of fast aggregation and higher levels of secondary nucleation (fig 2, fig S2 and S3), it is not saturated in our measured concentrations and the double logarithmic plot shows no kink in its linear correlation (**Fig. 1**).

4) Page 7, line 178 – are oligomers formed by rounds of freeze-drying physiologically relevant?

Response: We originally checked different conditions to get a better yield of oligomer formation as it is normally very low. Every round of α SN monomer freeze-drying in water leads to further accumulation of oligomers. Even though the reason of higher yield oligomer formation as the result of freeze drying remains unclear (we assume a high energy barrier for oligomer formation that freeze drying helps monomer to cross), this method does not alter oligomer structure as shown by SAXS (fig S6 C). Therefore, we decided to test the formation of oligomer especially for shorter mutants with three times freeze-drying, considering that the formation of α SN oligomer happens in very low yield and varies from batch to batch (1 to 5% of monomer); furthermore shorter mutants lacking C-terminal Tyr residues are not detected so well due to reduced UV absorption. To make this point clear in the manuscript, we added a short description in the oligomer part in result section: “To test the general tendency of forming α SOs, we first optimized oligomerization conditions for α SN-FL by exploring parameters such as NaCl concentration, pH, shaking, multiple rounds of freeze-drying as well as the α SO inducing agent 4-oxononenal (ONE). Among these, multiple rounds of freeze-drying 44 and ONE modification 33 were most effective.”

5) Page 7, line 181 – where in figure S5DE do the authors have the NaCl data? How does this differ to the PBS buffer (that I assume the other oligomers are in?). Buffer differences have been shown to be important for fibril polymorph formation, this could be interesting for oligomers too.

Response: Thank you for highlighting this. The NaCl data have now been added to Fig. S5 (now renumbered to Fig. S8). The concentrations of NaCl only vary in panel F and are now clearly indicated. All experiments are in PBS indicated otherwise.

6) Page 8, line 186 – The peak size for the aSOs in Figure S5G don't look to give the same yield for 121 and 125 compared to 130 and 135.

Response: We appreciate the need to clarify. For better comparison, we used the oligomer/monomer peaks height ratio. Even though we see different peak heights for oligomers, the oligomer/monomer peak ratio does not follow a specific trend for different mutants as can be seen in fig. S5 I (in new version: fig. S8 I.)

7) Page 8, line 188-189 – It's not clear why ONE was used or how this is aiding oligomer formation of shorter fragments or why this is useful to investigate.

Response: We have now added the following part to the discussion: “ONE is a product of peroxidation of polyunsaturated fatty acids present in the cell membrane² which has been reported to cross-link brain proteins³. In particular, ONE can induce α SN oligomer formation which could be physiologically relevant considering the role of oxidative stress in PD^{4,5}. Therefore, we investigated ONE-induced oligomer formation in CTD truncated mutants that do not form oligomers on their own.”

8) Page 9, line 216 – Is there a similar mechanism linking calcein release and cell toxicity? This seems a bit of a weak or presumptive link.

Response: We have now added the following at the beginning of the Results section on calcein assays: “There is evidence that permeabilization of cell membranes by α SOs leads to calcium ion influx and cell death⁶⁻⁸. As a simple model system to mimic this phenomenon, we determined the ability of these α SO constructs...”

9) Discussion – What are the most abundant truncated species identified in brain tissue? Are there species identified in this work that promote fibrillisation of FL aSN, cell toxicity and calcein release that have also been identified in vivo and may thus be more important to investigate further?

Response: Great point. In the Introduction we add that “Truncations at residues 103, 110, 113, 114, 115, 119, 122, 124, 125, 133, and 135 have been identified *in vivo*⁹. Those at 119 and 122 at most prevalent and occur at 20-25% abundance relative to full-length α SN.” This is followed up by adding

at the very end of the Discussion zone: “Several CTD truncated α SNs have been found in neural cells or Lewy bodies ^{10,11} Our work identifies a potential “danger zone” of particularly deleterious CTD truncations, namely a CTD truncation range from α SN 121 to α SN 125 which shows increased fibrillation compared to α SNFL while retaining the ability to form cytotoxic α SN oligomers on their own. In this context it is striking that the most common *in vivo* truncated forms of α SN are α SN119 and α SN122⁹. This highlights the deleterious consequences of biological processing of α SN.”

10) Page 12, line 281-294 – This section is not very clear, I’m not sure what the use of the ONE oligomers is showing, that you need N and C terminal interactions to form oligomers?

Response: We have tried to reformulate our thoughts to make it clearer and now provide the following section: “It is remarkable that ONE induces oligomer formation even in shorter constructs which are normally not able to form oligomer. Given that ONE-oligomers and unmodified oligomers have similar size and shape according to SAXS, we conclude that ONE helps otherwise unstable truncated oligomers form, instead of directing α SNs towards a different class of oligomers. In other words, ONE could be seen as filling in CTD’s role in stabilizing the oligomer. The basis for this remains unclear. One speculative possibility is that just as the CTD’s anionic tail may reduce the electrostatic repulsion caused by the more cationic N-terminal region, similarly ONE may overcome the N-terminal positive charges either by modifying the Lys side chains or by cross-linking them to other parts of the protein. If inter-molecular interactions between CTD and NTD in the oligomeric state indeed helps form and stabilize the oligomers, then conversely, shortening of the CTD will decrease the frequency and strength of these interactions, leading to extended tails and therefore oligomers with bigger shells. However, there is a limit to the ability of ONE to compensate for truncation and eventually removal of sufficient residues (around α SN115) completely abolishes the ability to form oligomers without help from ONE.”

11) Page 12, line 295-303 – Are changes in the charge of the protein or aSOs upon truncation likely to affect the interaction of α SN with the vesicles?

Response: To address this, we add the following to this section: “Only NTD and NAC regions are known to interact directly with vesicles in monomeric α SN ^{12,13}, but truncation of CTD may have indirect effects by altering the core-shell distribution within the oligomer.” We then link this up with a discussion of the oligomer-membrane interactions known from the literature.

12) Supplementary Page 1, line 24 – What ThT concentration was used?

Response: Thank you, this is now indicated. 40 μ M ThT was used for all experiments.

13) Supplementary Page 7 and 8 - Figure S2 and S3 could the authors please provide the number of replicates and wells used in the data.

Response: We have now added “Three replicates in three wells were used for every concentration” to the legends.

14) Supplementary Page 10 - Figure S5a-e – are these FL aSN?

Response: Yes, this is now clarified in the legend to what is now Fig. S8 (aSN-FL in panels A-F).

Reviewer #3 (Remarks to the Author):

In the paper entitled ‘The C-terminal tail of α -synuclein protects against aggregate replication but is critical for oligomerization’, the authors describe the results of a study aimed at determining effects of C-terminal truncations on the self-assembly properties of alpha-synuclein (aSyn), a protein that forms aggregates in the brains of individuals with Parkinson’s disease (PD) and other synucleinopathy disorders. The investigators report that aSyn truncation variants with 121 or fewer residues have a substantially increased propensity to form amyloid-like fibrils, apparently because they more readily undergo secondary fibrillization as a result of enhanced fragmentation. An increased rate of fibrillization is also observed in mixtures of full-length aSyn with as little as 10% (mol/mol) of the aSyn-110 truncation variant. In contrast, C-terminal truncation interferes with aSyn oligomerization, and oligomers formed by truncation variants in the presence of the pro-oxidant 4-oxo-2-nonenal have a reduced propensity to permeabilize membranes and elicit cytotoxicity. The authors attribute these effects of the aSyn C-terminal tail to its electrostatic interactions with the N-terminal and NAC (central hydrophobic) domains, which favor oligomerization but interfere with the formation of amyloid-like fibrils.

This paper is significant in terms of its relevance to molecular mechanisms involved in the pathogenesis of PD and other synucleinopathy disorders. In particular, the results are of a high impact because they yield important insights into structural features of the C-terminal domain of aSyn that modulate two key pathogenic events – the protein’s conversion to potentially neurotoxic oligomers and the formation of aSyn amyloid-like fibrils that can spread throughout the brain. The findings reported here are novel and represent an important advance in the aSyn field.

Strengths of the paper in addition to its significance include the fact that it is well written and organized, and the authors have been thorough in terms of referencing the key literature of relevance to this study. The experiments have been carefully executed, yielding high-quality results, and the data processing and mathematical modeling are rigorous. On the other hand, there are several errors in the text and figures/figure legends, and parts of the paper are unclear. In addition, some of the authors’ conclusions would be strengthened with additional discussion or by including additional data. These issues should be addressed in a revised version of the manuscript.

Response: We thank the reviewer for this generous assessment and for the constructive suggestions for improvement which we address below.

Specific comments:

1. p. 4, line 86, ‘is still lacking’; delete

Response: This phrase was removed.

2. p. 5, line 95, ‘the CTD (residues 101-140)’; previously the CTD was described as a segment spanning residues 96-140 (the same polypeptide segment should be referred to throughout the paper).

Response: This part was rephrased. Residues from 101 to 140 correspond to the region that we worked with in our CTD truncated versions.

3. p. 5, line 110, ‘high concentrations with a float slope’; replace ‘float’ with ‘flat’.

Response: Done.

4. p. 6, lines 123-125, ‘and the rate-limiting step shifts from monomer binding to the end of fibrils (which is dependent on [monomer]) to the conformational conversion of the monomer to the fibrillated state (which is not)’; it is unclear why the [monomer]-independent phase at high monomer concentrations is interpreted to reflect a conformational change of monomer to the fibrillated state (a component of the primary fibril elongation phase), as opposed to the secondary [monomer]-independent processes of ‘fragmentation or saturated secondary nucleation’ referred to earlier (lines 117-118). It would be helpful if this point were clarified.

Response: We clarified this statement by adding an explanation to the sentence. “Considering elongation process as a two-step reaction (*i.e.* binding of monomers to the end of fibrils and their conformational conversion to fibrils), saturation of elongation implies that at higher [monomer], the reaction no longer exhibits first order kinetics with respect to [monomer] and the rate-limiting step shifts from monomer binding to the end of fibrils (which is dependent on [monomer]) to the conformational conversion of the monomer to the fibrillated state (which is not)”.

5. p. 6, lines 138-139, ‘from plots of the initial rate of fibrillation against the total concentration of monomer and seed’; the plot in Fig. S4 shows only the seed concentration, not the total concentration of monomer and seed, on the x-axis (this point should be clarified).

Response: In the text we now replace total concentration with seed concentration, the only parameter that is varied in the experiment. As the concentration of monomer is constant (50 μ M), any changes in seed concentration reflect changes in total aSN concentration.

6. p. 6, lines 142-143, ‘To compare different constructs with different reaction orders of nucleation, we calculated the nucleation rate at 50 micromolar’; it would be helpful if a justification were provided for choosing 50 micromolar as the optimal concentration to account for different reaction orders.

Response: To calculate the reaction rates of nucleation from the composite rates of k_+k_n or k_+k_2 , we used the elongation rates which was obtained experimentally in 50 μM of monomer. Additionally, 50 μM is an intermediate concentration in the range of concentrations sampled, so a natural choice for a representative concentration. We added this reason in the text.

7. p. 6, line 153, ‘is around -0.5 for all mutants’; the value is closer to -0.3 to -0.4.

Response: for simplicity we mention the γ value -0.5 because theoretically this value indicates fragmentation or a saturated secondary nucleation. Smaller numbers show a probable saturation in elongation. We have now modified the text to read: “The scaling exponent γ ... is comparable to -0.5, the value obtained from theory for a system with fragmentation or saturated secondary nucleation, for all mutants”.

8. p. 7, line 158, ‘though an increase in the turnover rate on the surface of fibrils’; it would be helpful if this concept were explained in greater detail (turnover of what?)

Response: We thank the reviewer for prompting us to clarify this. We now clarify by adding “turnover rate, *i.e.* the rate of detachment of formed nuclei from the surface of fibrils and attachment of fresh monomer”.

9. p. 7, lines 162-163, ‘demonstrates the dominance of secondary processes which is most likely fragmentation’; additional detail should be provided to explain why the authors conclude that the stimulatory effect of agitation on the fibrillization rate reflects enhanced fibril fragmentation rather than turnover.

Response: We cannot unambiguously conclude that fragmentation is the only secondary process. Agitation has been shown to promote fragmentation in some systems, whereas its effect on increasing secondary nucleation may be reasonable but is, to our knowledge, not yet backed by experimental evidence in the context of amyloid formation. Thus, we argue in favor of fragmentation. We have now clarified this in the text where we have slightly toned down the emphasis on fragmentation to keep options more open and also emphasize that turnover increase is an option as well (though less likely).

10. p. 7, line 169, ‘60% aSN110 and higher showed the same behavior as 100% aSN110’; delete ‘and higher’ because 60% was the highest percentage of aSyn 110 tested in this experiment.

Response: Duly corrected.

11. p. 7, line 180, ‘without any change in structure according to Small Angle X-ray Scattering’; refer to Fig. S5C here.

Response: Duly corrected (now Fig. S8C).

12. p. 8, line 206, ‘and both aSOs and ONE-modified aSOs had diameters of approx. 10 nm’; it is unclear why the larger sizes of aSOs prepared from shorter C-terminal aSyn variants (determined via SAXS analysis) wasn’t evident in the TEM images. This point should be addressed in the discussion.

Response: Even though these two techniques give information about size of the oligomers, they have inherent differences in what they measure. While TEM is dependent on the staining and observations of the stained (or unstained in negative staining) objects, SAXS measures the scattering pattern of the molecules. Accordingly, we added the following statement to clarify this point: “Oligomer size measured by TEM however is similarly about 10 nm for all the mutants, arguing the core region was stained, while the fluffy shell was not observed by TEM.”

13. p. 8, lines 213-214, ‘ONE-modified aSOs of shorter constructs (ONE-aSO100 to ONE-aSO125) showed reduced disruption (Fig. 3IJ)’; replace ‘ONE-aSO125’ with ‘ONE-aSO121’.

Response: Duly corrected.

14. pp. 8-9, lines 214-215, ‘SH-CY5Y neuroblastoma cells’; replace ‘SCY5Y’ with ‘SH-SY5Y’.

Response: Duly corrected.

15. p. 10, lines 225-226, ‘Although elongation of CTD truncated α SNs was slowed down 4-fold in the shortest construct at 50 micromolar aSN due to saturation of elongation’; it would be helpful if the data supporting the ‘4-fold’ value in this statement were referred to here, as this is not evident in the data in Figs. 2A and S4.

Response: The 4-fold statement is no longer valid after calculating elongation rate taking the saturation into account. Therefore, we deleted the statement and added the relative elongation rate constants in the table in figure S6.

16. p. 10, line 240: reference 59 is missing from the bibliography.

Response: Thank you for noticing this. We added the reference.

17. p. 10, lines 248-251, ‘mutants containing disulfide bonds between residues 107 and 124 in CTD combined with disulfide bonds in the NTD-NAC region (residues 9, 42, 69 and 89) do not fibrillate; by contrast disulfide bonds in the NTD-NAC region alone are insufficient to inhibit fibrillation’; this statement should be reworded to indicate that intramolecular disulfide bonds linking cysteine residues introduced in the CTD and NTD-NAC regions were used in the study described in reference 51.

Response: Noted. For clarity, intramolecular disulfide bonds are highlighted in the track change version of the manuscript.

18. Fig. 2A-C: it should be explained why there are replicate points for some of the aSyn truncation variants (aSyn 105 in panels A and C; aSyn 125 and aSyn 130 in panels B and C; aSyn 135 in panel C).

Response: Each data point in panel A-C correspond to a separate experiment with three repeats in three wells; α SN 105, α SN 125, aSN 130, and α SN 135 were randomly repeated to strengthen the data. This is noted in the figure legend.

19. Fig. 2E-F: it should be made clear in the figure legend and in the corresponding part of the Results section that these experiments were carried out with full-length aSyn. Also, the paper would be strengthened by demonstrating that similar results are obtained when repeating these experiments with at least a subset of truncation variants (e.g. aSyn 125 and aSyn 130, to account for the pronounced difference in the secondary rate constants determined for these two variants in terms of a difference in the degree of fragmentation or secondary nucleation).

Response: α SN mutant was specified in the legend. These data are for α SN121; however, similar results were observed for the other mutants that are added as figure S7. This is mentioned in the text now.

20. Fig. 2G: it should be indicated on the graph that the ratios in the legend refer to ‘aSyn FL: aSyn 110’.

Response: Duly noted. We added a note to the legend indicating α SN FL:aSN 100.

21. Figs. 3, S6, S7 legends: the number of replicate data points and the method used to assess statistical significance should be indicated.

Response: Duly corrected.

22. Fig. 3 and Fig. S6H legends: the error bars should be defined (e.g. SD or SEM).

Response: Duly corrected.

23. Fig. 3D legend: specify that the number of monomers per oligomer for both kinds of oligomers increases with the extent of truncation.

Response: This is mentioned in the legend.

24. Fig. S2: it would be helpful to show the curve fits for aSyn 135, aSyn 130, and aSyn 125 for the secondary nucleation dominant and fragmentation dominant models separately, and to include MRE values on the plots as was done for aSyn WT. Also, the following statement in

the legend, ‘the models with saturation fit just as well’, should be clarified (e.g. refer to ‘the secondary nucleation dominant and fragmentation dominant models with saturation elongation, which produce a better fit with aSyn variants with more extensive C-terminal truncations (Fig. S3)’.

Response: We changed Fig. S2 accordingly and mentioned the saturation models in the legend.

25. Fig. S5C: the caption in the graph is difficult to read and should be enlarged.

Response: Duly corrected.

26. Fig. S5D, legend: the meaning of the asterisks written beside some of the pH values should be explained.

Response: Duly corrected.

27. Fig. S5F, legend: ‘between pH 6.5 and 9.5’ should be written as ‘between pH 6 and 9.5’.

Response: Duly corrected.

28. Fig. S5G, legend, ‘aSN 115 and shorter constructs’; delete ‘and shorter constructs’ because aSyn 115 was the shortest variant examined here.

Response: Duly corrected.

29. Fig. S6B: a single label at the top of the graph should be used to indicate that the curves correspond to a set of ONE-treated aSyn oligomers, rather than repeatedly inserting ‘ONE’ in all of the curve labels below (currently, the curve labels are difficult to see because of the crowded nature of the graph).

Response: Duly corrected.

30. Fig. S6D,F: the curve labels are difficult to see because of the crowded nature of the graph, and the term ‘ONE’ should be used at the top of the graph rather than repeatedly in the individual curve labels.

Response: Done. The legend also rearranged according to order of data from top to bottom for panel F.

31. Fig. S6G: correct the typo in the x-axis label.

Response: Noted. Thanks.

32. General comment about the Methods’ section: an additional subsection describing the methods used for statistical analysis should be included.

Response: Duly corrected. The averages and comparisons were mentioned in the statistical analysis subsection.

33. Supplement, p. 2, line 45, ‘Gaspar et al. [40]’; the Gaspar et al. reference is listed as reference 41 in the bibliography.

Response: We have corrected referencing throughout the manuscript. Note that we have separate referencing for main text and supplementary text.

34. Supplement, p. 3, line 61, ‘Oligomer Preparation’; additional detail explaining how the experiments represented by Fig. S5D-F should be provided.

Response: Noted. The following line was provided to explain the oligomer preparation in different salt or pH conditions: “To prepare the oligomers in different pH and salt concentrations, monomers were first dissolved in milliQ water and then diluted into a given working concentration with the desired buffers.”

35. Supplement, p. 3, lines 83-87, ‘Fibrils were sonicated to minimize light scattering and spectra were recorded from 250 to 185 nm in 1 mg/ml using 0.1 mm cuvette. All data are averages of three repeats with 0.5 nm steps per 0.5 s and 1 nm bandwidth. Thermal scans to monitor conformational changes upon vesicle binding were carried out as described. Briefly, a solution of 0.2 mg/mL monomeric aSN and 1 mg/mL DMPG was heated from 5 to 95 deg C at 1 deg C/min and ellipticity was monitored at 220 nm’; this text could be deleted because CD data are only shown for oligomeric aSyn in the absence of phospholipids (Fig. S7E,F).

Response: Duly corrected.

36. Supplement, p. 4, line 105, ‘or 0.4 mg/mL fibril’; delete.

Response: Duly corrected.

37. Supplement, p. 5, line 125, ‘different variants of N- terminal truncated’; this should read ‘C-terminal truncated’.

Response: Duly corrected.

38. Supplement, p. 5, line 127: presumably the streptomycin concentration was 100 micrograms per mL, not 100 mg/mL?

Response: Duly corrected.

REFERENCES

- 1 Lorenzen, N. *et al.* The role of stable α -synuclein oligomers in the molecular events underlying amyloid formation. *J. Am. Chem. Soc.* **136**, 3859-3868 (2014).
- 2 Lee, S. H. & Blair, I. A. Characterization of 4-Oxo-2-nonenal as a Novel Product of Lipid Peroxidation. *Chemical Research in Toxicology* **13**, 698-702, doi:10.1021/tx000101a (2000).

- 3 Stewart, B. J., Doorn, J. A. & Petersen, D. R. Residue-Specific Adduction of Tubulin by 4-Hydroxynonenal and 4-Oxononenal Causes Cross-Linking and Inhibits Polymerization. *Chemical Research in Toxicology* **20**, 1111-1119, doi:10.1021/tx700106v (2007).
- 4 Almandoz-Gil, L., Welander, H., Ihse, E., Khoonsari, P. E., Musunuri, S., Lendel, C., Sigvardson, J., Karlsson, M., Ingelsson, M., Kultima, K. & Bergström, J. Low molar excess of 4-oxo-2-nonenal and 4-hydroxy-2-nonenal promote oligomerization of alpha-synuclein through different pathways. *Free Radical Biology and Medicine* **110**, 421-431, doi:10.1016/j.freeradbiomed.2017.07.004 (2017).
- 5 Näsström, T., Fagerqvist, T., Barbu, M., Karlsson, M., Nikolajeff, F., Kasrayan, A., Ekberg, M., Lannfelt, L., Ingelsson, M. & Bergström, J. The lipid peroxidation products 4-oxo-2-nonenal and 4-hydroxy-2-nonenal promote the formation of α -synuclein oligomers with distinct biochemical, morphological, and functional properties. *Free Radical Biology and Medicine* **50**, 428-437, doi:10.1016/j.freeradbiomed.2010.11.027 (2011).
- 6 Danzer, K. M., Haasen, D., Karow, A. R., Moussaud, S., Habeck, M., Giese, A., Kretschmar, H., Hengerer, B. & Kostka, M. Different species of alpha-synuclein oligomers induce calcium influx and seeding. *J Neurosci* **27**, 9220-9232, doi:10.1523/JNEUROSCI.2617-07.2007 (2007).
- 7 Feng, L. R., Federoff, H. J., Vicini, S. & Maguire-Zeiss, K. A. Alpha-synuclein mediates alterations in membrane conductance: a potential role for alpha-synuclein oligomers in cell vulnerability. *Eur J Neurosci* **32**, 10-17, doi:10.1111/j.1460-9568.2010.07266.x (2010).
- 8 Flagmeier, P., De, S., Wirthensohn, D. C., Lee, S. F., Vincke, C., Muijldermans, S., Knowles, T. P. J., Gandhi, S., Dobson, C. M. & Klenerman, D. Ultrasensitive Measurement of Ca(2+) Influx into Lipid Vesicles Induced by Protein Aggregates. *Angew Chem Int Ed Engl* **56**, 7750-7754, doi:10.1002/anie.201700966 (2017).
- 9 Sorrentino, Z. A. & Giasson, B. I. The emerging role of α -synuclein truncation in aggregation and disease. *The Journal of biological chemistry* **295**, 10224-10244, doi:10.1074/jbc.REV120.011743 (2020).
- 10 Pieri, L., Chafey, P., Le Gall, M., Clary, G., Melki, R. & Redeker, V. Cellular response of human neuroblastoma cells to α -synuclein fibrils, the main constituent of Lewy bodies. *Biochimica et Biophysica Acta (BBA) - General Subjects* **1860**, 8-19, doi:<https://doi.org/10.1016/j.bbagen.2015.10.007> (2016).
- 11 Li, W., West, N., Colla, E., Pletnikova, O., Troncoso, J. C., Marsh, L., Dawson, T. M., Jäkälä, P., Hartmann, T., Price, D. L. & Lee, M. K. Aggregation promoting C-terminal truncation of α -synuclein is a normal cellular process and is enhanced by the familial Parkinson's disease-linked mutations. *Proceedings of the National Academy of Sciences of the United States of America* **102**, 2162-2167, doi:10.1073/pnas.0406976102 (2005).
- 12 Fusco, G., De Simone, A., Gopinath, T., Vostrikov, V., Vendruscolo, M., Dobson, C. M. & Veglia, G. Direct observation of the three regions in α -synuclein that determine its membrane-bound behaviour. *Nature Communications* **5**, 3827, doi:10.1038/ncomms4827 (2014).
- 13 Antonschmidt, L., Dervisoglu, R., Sant, V., Tekwani Movellan, K., Mey, I., Riedel, D., Steinem, C., Becker, S., Andreas, L. B. & Griesinger, C. Insights into the molecular mechanism of amyloid filament formation: Segmental folding of alpha-synuclein on lipid membranes. *Sci Adv* **7**, doi:10.1126/sciadv.abg2174 (2021).

REVIEWERS' COMMENTS:

Reviewer #1 (Remarks to the Author):

The authors adequately responded to my concerns, and the revised article is suitable for publication in the Communications Biology journal.

Reviewer #2 (Remarks to the Author):

The authors have satisfactorily addressed my comments. The manuscript is much clearer and has improved. I would recommend for publication.

Reviewer #3 (Remarks to the Author):

The authors have addressed the critiques from the first review satisfactorily and included additional data. As a result, the manuscript is now much improved. This paper will be of great interest to investigators in the alpha-synuclein field, as well as researchers interested in protein fibrillation mechanisms more generally.

There are only a few remaining issues to address:

(i) Fig. S8, legend: the lettering should be corrected ('G' is repeated).

(ii) Figs. S9 and S10, legend: the number of replicate data points and the method used to assess statistical significance should be indicated, and the error bars should be defined.

Reviewer #1 (Remarks to the Author):

The authors adequately responded to my concerns, and the revised article is suitable for publication in the Communications Biology journal.

Reviewer #2 (Remarks to the Author):

The authors have satisfactorily addressed my comments. The manuscript is much clearer and has improved. I would recommend for publication.

Reviewer #3 (Remarks to the Author):

The authors have addressed the critiques from the first review satisfactorily and included additional data. As a result, the manuscript is now much improved. This paper will be of great interest to investigators in the alpha-synuclein field, as well as researchers interested in protein fibrillation mechanisms more generally.

There are only a few remaining issues to address:

(i) Fig. S8, legend: the lettering should be corrected ('G' is repeated).

Response: The lettering is fixed now.

(ii) Figs. S9 and S10, legend: the number of replicate data points and the method used to assess statistical significance should be indicated, and the error bars should be defined.

Response: The information is provided now.

We are thankful for the valuable comments of the reviewers and believe that their comments improved the quality of our paper.